# Research on the impact of digital financial inclusion and environmental regulation on industrial structure upgrading-based on spatial Durbin model analysis

Liuwu Chen[1], Guimei Zhang[1,2]*

**1** School of KRIRK UNIVERSITY, Bangkok, Thailand, **2** School of Economics and Management, Sanming University, Sanming, China

* guimei@fjsmu.edu.cn

**Data Availability Statement:** All relevant data are within the manuscript and its Supporting Information files.

## Abstract

In the context of building a modern industrial system, digital finance inclusive (DFI) and environmental regulation (ER) have become significant drivers of industrial structure upgrading. Amid the rapid development of DFI and the tightening of ER, high-quality development goals have been set. From a spatial perspective, can DFI, ER, and their interactions positively impact China's industrial structure? Based on panel data from 30 provinces in China from 2011 to 2020, empirical analysis is conducted using the Spatial Durbin Model and the Threshold Model. The study find that, firstly, in spatial regression, both DFI and ER significantly promote the upgrading of industrial structures, with notable spatial spillover effects. Secondly, it was discovered that the interaction between DFI and ER have a significant negative impact on industrial structure upgrading, with a pronounced negative spatial spillover effect, indicating a "siphon effect," which is a new finding. In the threshold model regression, it is found that both DFI and ER exhibit a single-threshold effect, while their interaction shows a double-threshold effect. In the heterogeneity test, it is found that the interaction between DFI and ER on industrial structure upgrading is greatly influenced by regional factors, with significant differences across different regions. Therefore, it is crucial to pay attention to the threshold values in the threshold effect while considering the unique economic and geographical attributes of each region to formulate targeted strategies.

## 1. Introduction

At present, China's economy is experiencing a "structural slowdown." The negative impacts of disorderly expansionist investment and production methods [1, 2] have severely hindered China's economic development and the upgrading and transformation of its industrial structure. China is fully aware of the harm caused by resource-intensive and extensive development models and has abandoned such growth patterns. In the report of the 20th National Congress, the Chinese Communist Party emphasized the importance and urgency of developing a

**Funding:** (1) the Natural Science Foundation of Fujian Province (Grant Number: 2020JOI927, Recipient: Guimei Zhang) (2) the Social Science Foundation of Fujian Province (Grant Number: FJ2023X006, Recipient: Guimei Zhang). The funders had no role in study design, data collection and analysis, decision to publish, or preparation of the manuscript.

**Competing interests:** The authors have declared that no competing interests exist.

modern industrial system. To achieve the strategic goals of carbon peak and carbon neutrality by 2030 and 2060, China is actively exploring a high-quality development path that aligns with its unique characteristics and development needs.

Capital is an indispensable core element in the process of industrial transformation, with finance serving as the heart of capital allocation. DFI, as a new form of inclusive finance characterized by "low barriers to entry, wide coverage, and high efficiency," allocates resources more efficiently to sectors with strong growth potential and higher added value. It significantly reduces operational costs for small and medium-sized enterprises (SMEs), mitigates financial exclusion caused by informational asymmetries, lowers financing thresholds for SMEs, and expands their financing channels. This alleviation of financing difficulties and high costs for SMEs further unlocks their production potential, enhancing their capacity for sustainable development, innovation, and research and development. This optimizes the industrial structure and promotes the upgrade of China's industrial framework.

ER, as a strategy for addressing corporate pollution and environmental degradation, employs governmental administrative measures to either penalize corporations or internalize the costs of emissions. This results in increased operational expenses for companies, potentially leading to bankruptcy. In response, businesses are compelled to innovate, enhancing the technological sophistication of their products to reduce pollution; alternatively, they may relocate production to circumvent environmental regulations. ER facilitates the redirection of resources from industries traditionally characterized by "three highs and one low" industries to emerging sectors that are energy-efficient, technologically advanced, and have higher added values. This process is instrumental in advancing the structural upgrade of China's industry.

Based on this, this paper conducts an in-depth analysis of the relationship between DFI and industrial structure upgrading from both theoretical and empirical perspectives within a spatial framework. It also examines the relationship between ER and industrial structure upgrading, as well as the interaction between the two and its impact on industrial structure upgrading. Additionally, regional heterogeneity analysis is performed. In the threshold model, the study analyzes whether there are threshold effects of digital inclusive finance, environmental regulation, and their interaction on industrial structure upgrading. This will help consider the unique economic and geographical attributes of each region's impact on industrial structure upgrading and provide policy recommendations.

This paper's potential marginal contributions are as follows: First, it offers an innovative research perspective. Most existing studies adopt an independent spatial perspective. DFI and ER have spatial spillover effects that can impact surrounding areas and influence corporate decisions. This paper innovatively integrates DFI, ER, and industrial structure upgrading into a unified spatial econometric framework for analysis. Second, it presents an innovation in mechanism analysis. It refines theoretical mechanism analysis by examining from multiple perspectives the direct and indirect spatial effects of DFI, ER, and their interaction on industrial structure upgrading. Research indicates that the industrial structure is greatly influenced by regional factors, with significant variations in effects across different regions. Policies need to fully consider the unique economic and geographical attributes of each region.

## 2. Literature review

### 2.1. Research on DFI and industrial structure upgrading

When studying the impact of DFI on the upgrading of industrial structures, most scholars employ linear or nonlinear mathematical models for their research.

**2.1.1. A linear study of DFI to promote industrial structure upgrading.** Levine [3] argued that financial development was a key force driving the upgrade of industrial structures.

Goldsmith [4] highlighted that financial development significantly promoted the upgrading of industrial structures. Furthermore, Bruhn and Love [5] suggested that inclusive finance, through expanding financial service coverage, optimized the allocation of funds across industries, thereby facilitating the upgrading of industrial structures. Du et al. [6] used a mediation model to demonstrate that DFI significantly contributes to the optimization and upgrading of industrial structures. The study also identified consumer demand, technological innovation, income, and capital accumulation as mediating variables that further promote industrial structure upgrading. Using a spatial econometric model, Ren et al. [7] found that digital finance effectively promotes industrial structure upgrading, with the coverage breadth of digital finance being the most significant sub-indicator.

**2.1.2. A study of nonlinear relationships.**   Tang et al. [8], through an empirical study using panel data from 2011 to 2015, determined that a nonlinear relationship exists between DFI and the advancement of industrial structures. Duan and Li [9] empirically concluded that a nonlinear, inverted 'U-shaped' relationship exists between DFI and industrial structure upgrading, showing a threshold effect through a static panel model. Zheng and Guo [10] used a threshold model for their empirical research and discovered that the presumed nonlinear relationship between digitization and the rationalization of industrial structure does not hold. Instead, they found a nonlinear relationship between two key aspects of DFI—its breadth of coverage and depth of use—and the process of upgrading industrial structures. Guo and Jin [11] used the spatial Durbin model(SDM) to examine the nonlinear relationship between DFI and industrial structure upgrading. Their study revealed an inverted 'U-shaped' relationship characterized by a threshold effect

**2.1.3. Regional heterogeneity study.**   Cao and Zhou [12] analyzed the regional heterogeneity between DFI and industrial structure upgrading using provincial panel data. They noted that the impact is more significant in the eastern region compared to the western region. Wang [13] analyzed county panel data and determined that the impact of DFI on industrial structure upgrading is significantly more pronounced in the eastern and central regions. Li et al. [14] used urban panel data to analyze and conclude that there is significant urban heterogeneity between DFI and industrial structure upgrading. Liu et al. [15], utilizing prefecture and city panel data, identified that the influence of DFI on industrial structure upgrading is weaker in three types of cities: western cities, less economically developed cities, and resource-dependent cities. Ren et al. [7] conducted an analysis of heterogeneity, revealing that digital finance has a more significant effect in cities with robust economic development, lower levels of financialization, and smaller income gaps between urban and rural areas. [16] DFI plays a limited role in advancing the advancement of industrial structures in midstream regions, areas directly traversed by major routes, and provincial capital cities.

## 2.2. ER and industrial structure upgrading research

The recent focus on "green development" has led to stronger ER. Scholars have been studying the relationship between ER and industrial structure upgrading. Their conclusions fall into three categories: promotive effect, suppressive or minimal effect, and nonlinear relationship.

**2.2.1. ER promotes the upgrading of industrial structure.**   Qiu et al. [17] found that ER effectively drove the upgrading of industrial structures. Domazficky and Weber [18] argued that ER positively impacts corporate innovation, ultimately facilitating the continuous optimization of industrial structures. Lin and Guan [19] constructed spatial weight matrices based on geographic and economic characteristics and used the SDM model to study the relationship between ER and regional manufacturing upgrading in 30 provinces and cities in China. It was found that ER can provide support for manufacturing upgrading. Zeng and Liu [20] found

that the intensity of ER does not directly promote or inhibit the transformation and upgrading of the local industrial structure, but it has a positive spatial spillover effect on the upgrading of industrial structures in neighboring areas. Yan et al. [21] conducted an empirical analysis using the SDM to examine the impact of environmental regulation on industrial structure upgrading. They found a significant positive spatial correlation in industrial structure upgrading. In the national sample, environmental regulation aids in industrial structure upgrading, but regional differences exist. Yu and Zheng [22] used panel data from 128 prefecture-level cities in the Yangtze River Economic Belt, covering the period from 2000 to 2020, and performed empirical analysis with the SDM. The results show that formal environmental regulation has a significant positive spatial effect on the rationalization and upgrading of industrial structures.

**2.2.2. Environmental regulation has inhibitory effect or limited effect on industrial structure upgrading.** Xie and Guo [23], along with Brunnermeier and Cohen [24], argued that using ER to facilitate industrial structure upgrading fundamentally relies on technological innovation. They further contend that exclusively relying on environmental regulation to achieve industrial structure upgrading is not sustainable. Gao [25] conducted an empirical analysis using the SDM and found that investment based ER has an inhibitory effect on the upgrading of industrial structures in nearby areas. Local governments often compete to attract liquidity and achieve rapid economic growth, frequently by weakening environmental regulations [26–28]. However, this practice, according to the findings, is not favorable for optimizing and upgrading the regional industrial structure. The findings show a negative relationship between ER and industrial restructuring, suggesting that strict ER could hinder industrial adaptation and transformation. Blackman et al. [29] found that when Mexican environmental regulatory agencies increased the intensity of ER, there was no significant change in the environmental performance of polluting enterprises. Yu and Wang [30] found that the impact coefficient of environmental regulation on industrial structure upgrading is very small, indicating that its effect is limited. Zhang et al. [31] conducted a spatial econometric analysis of environmental policies in the Beijing-Tianjin-Hebei region of China. They found that the ER in these three regions do not influence each other, but the industrial structure upgrading exhibits significant negative spatial dependence.

**2.2.3. Non-linear research.** Guan and Qu [32], through spatial empirical analysis using panel data at local and municipal levels, found that ER exhibits a threshold impact on industrial structure upgrading, characterized by a "U-shaped" relationship. Liang and Li [33] used a threshold model to reveal the dual threshold effect of ER on industrial structure upgrading. Sun et al. [34] found an inverted 'U-shaped' correlation in the enhancement of industrial structures when assessing the effects of three environmental regulation tools: administrative orders, market incentives, and public participation. Yuan and Xie [35] investigated the threshold properties of ER's impact on industrial restructuring using a threshold regression model. Hu and Chanfan (2024) found that the impact of various ERs on industrial upgrading exhibits a clear "double threshold effect" through its impact on land use efficiency. He and Zheng [22] discovered through SDM that environmental regulation had an inverted U-shaped direct impact on rationalization and an U-shaped direct impact on progress. It also has an inverted U-shaped indirect impact on the rationalization and development of neighboring regions.

## 2.3. Literature summary

Although the existing literature is growing, current academic research on ER, DFI, and industrial structure upgrading remains fragmented. Most studies focus on a single aspect and do not systematically explore the relationship between DFI, ER, and industrial upgrading within a unified framework. Furthermore, there is a lack of theoretical discussion on how ER and DFI

can promote industrial upgrading. Existing studies also overlook the impact of the interaction between ER and DFI on industrial upgrading under spatial analysis.

The organization of this study is as follows: Section 2 reviews the related literature. Section 3 presents the theoretical analysis and hypotheses. Section 4 constructs the empirical model, including the SDM, regional heterogeneity test, threshold analysis, robustness test, and analysis of the empirical results. Section 5 provides conclusions and policy recommendations. Section 6 discusses insights and shortcomings.

# 3. Rational analysis and theoretical hypotheses

## 3.1 Mechanisms of DFI on industrial structural upgrading

**3.1.1. DFI upgrades the industrial structure of the region.** DFI plays a major role in industrial structure upgrading by promoting resource allocation. First, enterprise financing drives swift industrial advancement through the efficient allocation of credit resources, which in turn facilitates industrial structure upgrading [36, 37]. DFI is characterized by low thresholds, high coverage, and high efficiency, driven by technology empowerment through big data and artificial intelligence. It can accurately identify enterprises with development potential among numerous small, medium, and micro enterprises. Additionally, it quickly discovers new business forms and high-return opportunities in the market, thereby accelerating capital flow efficiency. This enhances financial efficiency and the availability of credit for MSMEs, eases capital financing difficulties, and promotes the growth of new business forms and enterprises with development potential. Therefore, DFI can promote the development of new industries and the upgrading of industrial structures through the efficient allocation of credit capital. Secondly, Jin [38] believed that DFI could promote the development of MSMEs by addressing information asymmetry, thereby promoting industrial structure upgrading. MSMEs frequently face issues like opaque operations, financial non-disclosure, limited scale, and lack of information transparency. Consequently, during quality assessment, these enterprises often experience inadequate evaluations due to insufficient information. DFI leverages big data platforms for financial innovation to create supply chain finance. This enables effective collection, data analysis, and real-time monitoring of MSMEs' information, which can effectively mitigate the financial exclusion caused by information asymmetry. Thus, DFI provides high-quality financial services to MSMEs, promoting their growth and strength. Thirdly, DFI creates online services, avoiding the high rent costs of offline outlets and reducing financial companies' operating costs. Online transactions also break regional restrictions, offering a wider business reach. Big data empowerment strengthens risk management, reduces credit risk, and minimizes financial institutions' potential losses. This reduction in capital allocation and transaction costs promotes the transfer of resources to higher growth and value-added sectors, facilitating the upgrading of the industrial structure and promoting the development of the financial sector.

DFI works in upgrading industrial structure through technological innovation. Ngai and Pissaride [39] believed that technological innovation is a crucial component industrial optimization and upgrading. Whereas the scarcity of traditional financial resources can significantly impede the development of innovative activities. [40]. Therefore DFI provides complementary support for technological innovation in SMEs Bai et al. [41]. For SMEs transitioning from low-end to high-end processes, technological innovation, undoubtedly facing the problem of long cycle, high cost, high risk, so it is indispensable to financial support, DFI as a product of science and technology finance can do "precise drip irrigation", on the one hand, DFI will make financial institutions increase the scale of investment in science and technology, specialized new and new industries, solve the problem of enterprise financing difficulties, financing

expensive problems. On one side, DFI will enable financial institutions to increase the scale of investment in science and technology-based, specialized, new and new industries, address the problem of challenges related to difficult and costly financing for businesses, and provide sufficient funds to accelerate the development of emerging industries and facilitate the upgrading of industrial structure. On the other side, DFI itself is the result of technological innovation, which helps the growth of digital technology, realizes the sharing of information resources, reduces the threshold and cost of access to information resources, helps the implementation of new technologies, further accelerates t the tailored development of DFI services according to local conditions, aids enterprise development, and promotes the upgrading of industrial structure.

DFI drives the upgrading of industrial structure by meeting consumption demand. Zhang et al. [42] discovered through their analysis that enhancing consumer demand acts as the major role in driving the upgrading of industrial structure. Firstly, there is a "long-tail group" in the financial market, and DFI provides financial services to these groups, increasing their educational opportunities and meeting their needs. This improvement in personal quality enhances human capital, generates higher labor productivity, aids enterprise innovation, and promotes industrial structure upgrading. Improving personal quality enhances human capital, resulting in higher labor productivity, fostering enterprise innovation, and promoting industrial structure upgrading. Secondly, the comprehensive nature of DFI can boost residents' income levels, especially benefiting low-income groups [43]. It also reduces income disparity between urban and rural regions and increases the demand for consumer credit. According to Keynesian theory, consumer demand increases with higher personal income and greater consumption expectations. This shift moves consumption from low-grade to middle- and high-grade, increasing development and enjoyment consumption, impacting the product market, and demanding more sustainable production, which influences the industry structure [44]. DFI will reduce spending on survival necessities, optimizing family consumption structures. This consumption upgrade promotes resource flow to medium- and high-end industries, boosts the production of mid-range and high-end products, and facilitates industrial structure advancement.

**3.1.2. DFI promotes the upgrading of the industrial structure of neighboring regions.** By improving the efficiency of financial services, DFI contributes to the advancement of the industrial structure in adjacent regions. On one hand, traditional financial institutions are usually concentrated in urban and economically developed areas. In contrast, DFI, driven by the large-scale construction of financial infrastructure and using Internet and mobile communication technology, overcomes geographic constraints, expands coverage, and improves financial accessibility [45]. This greatly enhances financial services in rural areas, townships, and adjacent regions. Even remote and marginal areas can enjoy similar financial services to those in urban areas, achieving "basic financial services without leaving the village, and comprehensive financial services without leaving the town," and reducing the density of geographic distribution of financial services. On the other hand, improving financial infrastructure promotes mutual exchanges and information-sharing among neighboring regions and the adoption of advanced technologies. This brings more opportunities for innovation and entrepreneurship to these regions, driving industrial structure upgrading.

DFI aids in enhancing the industrial structure of nearby areas by reducing the costs associated with financial services. Traditional financial services often require the establishment of physical outlets and manual services, which are costly. Conversely, DFI utilizes digital technology to remove geographical barriers, offering the benefits of low thresholds and reduced costs. This enables individuals in certain regions to access financial services online [46, 47]. Such an approach substantially decreases the cost of financial services, minimizes the need for

intermediary steps and human resources, and enhances service efficiency. Consequently, businesses and individuals in less developed areas can access these services at a reduced cost. This enhances their competitiveness within the industrial chain and contributes to the upgrading of the industrial structure.

DFI promotes the upgrading of industrial structure in adjacent areas by stimulating innovation and entrepreneurship. Digital finance can create additional entrepreneurial opportunities by expanding market size, facilitating knowledge spillovers, and altering the mix of factors, as outlined by Baker. This is especially true in technology and innovation-heavy industries, thereby enhancing innovation and entrepreneurship in urban areas [48]. From the perspective of financial factor flow, finance, as a factor resource, possesses both liquidity and profit-seeking nature. When high-return emerging industries exist in neighboring regions, DFI will flow to those areas, providing convenient financing channels and financial services to support the development and growth of SMEs. This will stimulate entrepreneurial and innovative vitality in the adjacent region, promote the growth of industrial structures towards technology- and knowledge-intensive sectors, and support the development of the local industry.

Therefore, this study proposes hypothesis 1: DFI is expected to exert a direct positive influence on the enhancement of the local industrial structure, and have a beneficial spillover effect on adjacent regions.

## 3.2. Theoretical analysis of the influence of ER on industrial structure upgrading

**3.2.1. The impact of ER on the upgrading of industrial structure in the region.** The function of capital and technical barriers of ER. In a specific industry, new enterprises must purchase extensive pollution control equipment before beginning operations to meet mandatory ER requirements on pollution indicators. This raises their pre-investment costs and increases operating expenses. The increased capital requirements for potential enterprises to enter a specific industry create capital barriers, which hinder the entry of new enterprises to some extent. Simultaneously, government departments enforce strict environmental screening conditions for enterprises entering a specific industry. To enter, enterprises must meet ERs regarding equipment, technology, processes, and other innovations. These regulations create technical standards that act as barriers, excluding backward enterprises from the industry. ERs, through their capital and technical requirements, prevent market entry for enterprises with limited capital, outdated production methods, and backward production capacity. Concurrently, these regulations stimulate ongoing technological innovations within enterprises as they strive to comply with increasingly stringent environmental standards. This process ultimately enhances the region's industrial structure.

The transfer and replacement of ERs. From one perspective, in line with the "Pollution Haven" hypothesis, stricter ERs escalate operational expenses in heavily polluting industries. To reduce these costs and avoid expenses for sewage treatment, some enterprises may relocate from areas enforcing stringent ERs to those with more relaxed standards. This relocation can result in these regions becoming hubs for "three highs and one low" industries. The migration of such businesses out of the region paves the way for more technologically advanced and environmentally friendly industries, particularly in the service sector. Consequently, this shift enhances and optimizes the region's industrial structure. From another perspective, due to ERs, the government will ban inefficient, high energy consumption, and high pollution business sectors, reducing investment in these areas. As a result, the movement of production factors towards these sectors diminishes, increasing the costs of production and operation for businesses, causing some enterprises to withdraw from the market. Instead, these excess

production factors flow to more efficient sectors. This substitution directs factor resources to industries represented by the service sector, contributing to the development of tertiary industries and the upgrading of the industrial structure.

Innovation compensation from ER. Porter's hypothesis states that while increased ER initially raises costs, in the long run, firms can reduce marginal pollution control costs by improving technological innovation and productivity, thereby achieving innovation compensation from ER. Brunnermeier and Cohen [24] suggested that environmental regulation could increase research and development investment in the manufacturing industry and promote technological innovation. When the industrial sector faces tightening ER, enterprises must choose between closure or transformation and upgrading. In the short term, this increases operating costs, especially environmental governance costs, which is not conducive to enterprise development and leads to the bankruptcy of backward enterprises, removing them from market competition. Over the long term, ER encourages companies to enhance their capacity for independent innovation and learning. This not only leads to long-term cost reductions and efficiency improvements but also fosters innovation through technological advancements and creative breakthroughs. ER promotes the enhancement of enterprises' independent innovation and learning abilities. This results in long-term cost reduction and efficiency improvement, elevates the production process, increases product added value through technological innovation, expands market share, and boosts the competitive edge of businesses. The development of enterprises and their technological innovation capabilities form a positive interactive mechanism, continuously consolidating this "win-win" pattern, greatly boosting the upgrading of industrial structures.

The demonstration effect and industrial synergy created by ER in neighboring areas will facilitate the advancement of the industrial structure in those regions. Firstly, the upgrading of the industrial structure, technological enhancement, and economic growth resulting from a positive ecological environment in a region will set a commendable example for neighboring areas [49]. Businesses in these adjacent regions tend to observe, learn, and emulate those in areas with advanced industrial structures, acquiring sophisticated technologies. This phenomenon, known as the 'free-rider' effect [50], facilitates the improvement of industrial structures in these neighboring regions, thereby promoting their industrial upgrade. Secondly, ER imposing stipulations on pollution and resource usage by local businesses also impact the supply chain, affecting components, intermediates, or services. Upstream and downstream partner enterprises in adjacent areas will need to offer more eco-friendly and superior quality products and services to meet these environmental standards. This necessitates a greater focus on training and recruiting talent in environmental protection technology, as well as escalating research, development, and innovation in this field. Consequently, businesses in neighboring regions will gravitate towards environmentally sustainable industrial practices and focus on advancing environmental protection technologies.

Therefore, this study proposes Hypothesis 2: ER is expected to exert a direct positive influence on the enhancement of the local industrial structure and have a beneficial spillover effect on adjacent regions.

## 3.3.Theoretical analysis of DFI, ER and industrial structure upgrading

### 3.3.1. DFI, ER to promote the upgrading of industrial structure in the region.
Firstly, in scenarios of escalating ER, areas with advanced financial development face increased operational pressures for certain business sectors, leading to financial disparities. However, the presence of DFI concentrates on the long-tail segments, enabling owners of SMEs to benefit from improved financial resource allocation. This reduces enterprise financing costs, curtails rent-

seeking opportunities, and mitigates financial exclusion due to information discrepancies. By effectively lowering the financing threshold for SMEs, it provides financial backing for innovation, encourages the adoption of environmental and technological advancements, and aids in the upgrading of industrial structures. Secondly, DFI can help enterprises understand environmental regulatory requirements and impacts, and assess environmental risks. This enables enterprises to adjust their business strategies and respond proactively to changes brought about by environmental regulations. Thirdly, DFI provides financial resources and support to environmental and social entrepreneurs, fostering innovation and entrepreneurship in environmental protection and sustainable development sectors. This stimulates entrepreneurial vigor and social enterprises, facilitating the transformation of industries into environmentally friendly and socially responsible sectors. Fourthly, DFI supports enterprises in establishing a green supply chain and engaging in financial cooperation with various links in the supply chain. Through the DFI platform, enterprises can establish cooperative relationships with environmental certification organizations and green suppliers, steering the industrial structure towards a more sustainable and environmentally friendly trajectory.

**3.3.2. DFI and ER promote the upgrading of industrial structure in neighboring regions.** Firstly, from the perspective of innovation and technology transfer, DFI supports enterprises' environmental innovation and technology application under ER. These innovations and technology transfers can generate spillover effects, facilitating the advancement of industrial structures in adjacent regions. Enterprises share environmental management experience and technical knowledge through the DFI platform. This cooperation promotes interregional technology transfer and experience sharing, and the demonstration effect in neighboring regions can inspire other enterprises to adopt similar measures, thus benefiting from environmental management and industrial upgrading. Secondly, from the lens of industrial alliances and partnerships, DFI encourages the establishment of cooperative relationships and industry alliances. Under ER, enterprises form partnerships through DFI platform to jointly promote environmental protection programs and industrial structure upgrading. Such partnerships and alliances can expand the scope of influence, spread environmental protection concepts and practices to neighboring regions, and promote the upgrading and transformation of neighboring industrial structures.

Therefore, this paper proposes Hypothesis 3: ER and DFI have a positive moderating effect on the metropolitan region and neighboring regions.

**3.3.3. DFI, ER inhibit structural upgrading of industries in the region.** Initially, considering the aspect of risk, enterprises heavily influenced by environmental regulations typically fall into the category of "three highs and one low". These businesses often possess outdated production capabilities, requiring substantial investment in environmental protection infrastructure, pollution management, and resource optimization. Consequently, they face an elevated risk of market elimination. Loans extended by digital financial institutions to such enterprises might encounter high risks and costs, potentially resulting in increased interest rates. High interest rates may make loans unaffordable for these firms, thereby limiting their ability to undertake industrial structural upgrading. Changes and uncertainties in environmental regulations become significant risks for firms. Firms need to adapt to changing environmental standards and requirements by adjusting their production processes, changing their product mix, or finding alternative resources. Such uncertainties and risks make firms take a conservative approach to industrial structure upgrading, fearing they will not see a return on investment or face higher business risks. Secondly, in terms of prioritizing short-term benefits and returns, DFI may focus more on short-term gains, neglecting long-term sustainability considerations. This leads them to favor traditional industries or projects with higher short-term returns in capital allocation and support, while paying insufficient attention

to industrial structure upgrading projects with higher environmental requirements or longer return cycles. Thirdly, DFI is a type of financial innovation, and it is difficult to avoid financial "mission drift." The aberrant development of finance and excessive speculation lead to "de-realization to virtualization" and other issues [51], which can harm the enhancement of the industrial framework and hinder industrial structure upgrading. Additionally, there is a mutual substitutability between financial development and environmental regulation, which can weaken the function of advancing the industrial structure [52].

**3.3.4. DFI and ER inhibit the upgrading of industrial structure in neighboring regions.**   Firstly, from the standpoint of industrial transfer, ER can cause "three highs and one low" industries to relocate to neighboring areas. Government departments may adopt a "race to the bottom" strategy, introducing high-energy-consuming enterprises for GDP performance, which hinders industrial structure progress in adjacent regions. This strategy of introducing high-energy-consuming enterprises is not conducive to upgrading the industrial structure of neighboring regions. Secondly, from the perspective of knowledge and technology spillover, strict ER require enterprises to introduce and apply new environmental protection technologies and techniques, which may involve difficulties in technology transfer and knowledge conversion. If DFI does not offer adequate technical support and knowledge exchange, enterprises in neighboring areas might face challenges like technology shortages and elevated costs of technology transfer. This can hinder their industrial structure advancement and negatively impact the growth of related upstream and downstream companies in these regions. Moreover, these upstream and downstream firms might face bankruptcy risks, further obstructing the enhancement of the industrial structure. Thirdly, considering the disparity in the availability of financial services, despite DFI's goal of providing widespread financial access, some regions still experience low penetration of these services. If enterprises and individuals in neighboring regions do not have full access to DFI, it will limit the financing and financial support they need to upgrade their industrial structure.

Therefore, this paper proposes 4:ER and DFI have negative moderating effects on the structural upgrading of industries in local and neighboring regions.

In summary, Fig 1 outlines the mechanisms through which digital inclusive finance affects the industrial structure upgrading in both local and neighboring regions. It also depicts the mechanisms of how environmental regulation influences the industrial structure upgrading in these regions, as well as the combined impact of their interaction on industrial structure upgrading. For details, see Fig 1.

## 4. Empirical analysis

**4.1. Measurement modeling.**   Industrial structure upgrading exhibits spatial correlation and dependence, as identified by Cheng et al. [53] and Zhang et al. [54]. These characteristics influence both the region in question and its neighboring areas. Traditional econometric models fall short in adequately explaining the spatial spillover effect, leading to significant errors in estimation results. Therefore, employing a spatial econometric model for analysis is more appropriate. The model, as formulated by Li et al. [55], is designed as follows:

$$A_{it} = pW*A_{it} + \beta_1 ER_{it} + \beta_2 DFI_{it} + \beta_3 ER_{it}*DFI_{it} + \beta_3 W*ER_{it} + \beta_4 W*DFI_{it} + \beta_5 W*ER_{it}*DFI_{it} + \gamma Controls_{it} + \varepsilon_{it} + V_{it} + u_i \tag{1}$$

Where i denotes province, t denotes year, www is the spatial matrix (this study selects the economic geography matrix for empirical analysis and chooses provincial distance as the robustness test matrix), A is the industrial structure advancement, which measures the upgrading of the industrial structure; DFI is the digital financial inclusion index, reflecting the level of

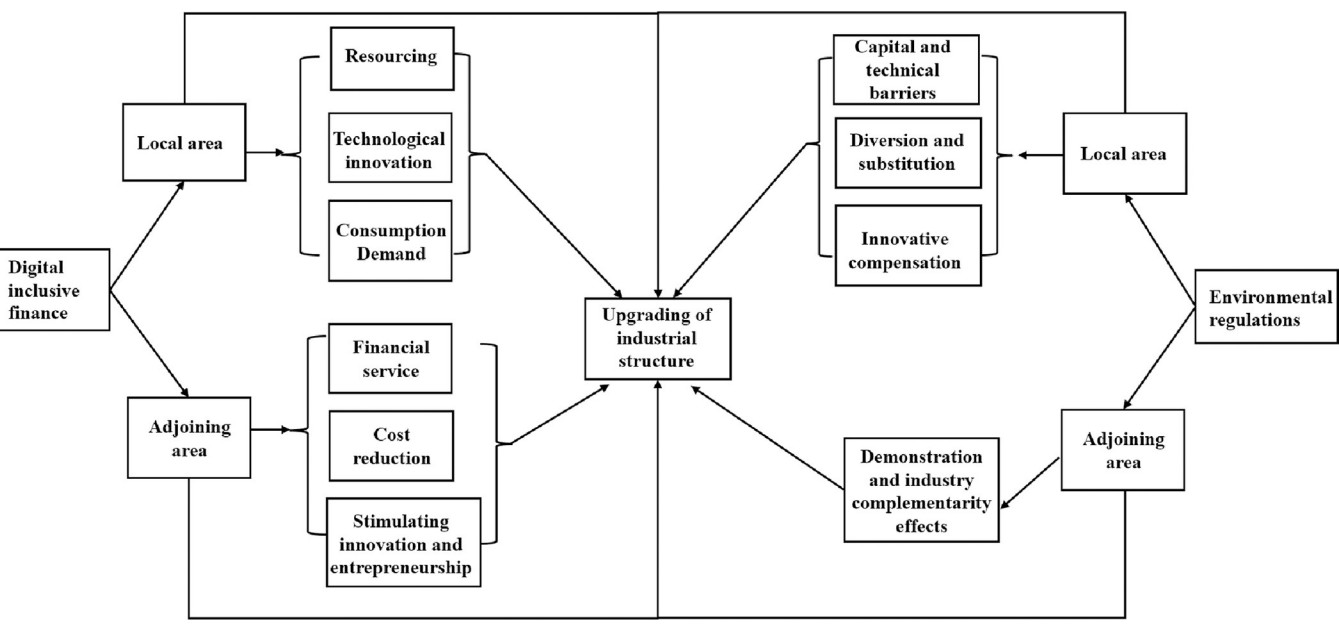

**Fig 1. Mechanism of action diagram.**

DFI in each province; ER is the strength of environmental regulation, reflecting the level of regulation in each province; Controls denotes the control variables; ε is the error term; u is the time fixed effect; and v is the individual fixed effect.

## 4.2. Variables and data descriptions

Explained variable: industrial structure upgrading (A). As the economy progresses, the shares of both secondary and tertiary industries rise. In this study, following Gan et al. [56], we selected the index of advanced industrial structure, specifically the ratio of the tertiary industry to the secondary industry, to measure industrial structure upgrading. A higher ratio indicates a higher level of industrial structure enhancement.

For explanatory variables, particularly ER, this study adopts the approach of Shangguan and Ge [57]. This method includes utilizing metrics such as the rate of industrial sulfur dioxide removal, the compliance rate of industrial wastewater discharge, and the comprehensive utilization rate of industrial solid waste. These are integrated using the entropy value method to quantify the intensity of ER, where a higher index signifies stronger ER. The process of finding the weight of each indicator value is:

(1) Calculate the weight of the indicator value for the ith item under the jth indicator:

$$p_{ij} = r_{ij} \bigg/ \sum_{i=1}^{m} r_{ij} \tag{2}$$

Calculate the entropy value of the jth indicator Ej:

$$e_j = -k \sum_{i=1}^{m} p_{ij} \cdot \ln p_{ij} \tag{3}$$

Among them:

$$k = 1/\ln m \tag{4}$$

(3) Calculate the entropy weight of the jth indicator:

$$W_j = (1 - ej)/\sum_{j=1}^{n}(1 - ej) \tag{5}$$

DFI, this study refers to Zhang et al. [58] using Peking University's Center for Digital Inclusion Finance to compile an index as a measure, the greater the index of DFI indicates the more rapid the development of the data source Peking University's Center for Digital Inclusion Finance.

Control variables:

(1) the level of government intervention (GOVER). The ratio of each province's fiscal expenditure to its GDP is used as a measurement [58]. Fiscal expenditure is chosen as a control variable because it significantly impacts environmental policies and industrial structure.

(2) an economic development level (PGDP). Per capita GDP of each province is used as a measurement [59]. The differing levels of economic development across provinces may lead to variations in the policies adopted.

(3) fixed investment level (FIX). The ratio of each province's fixed asset investment to its GDP is used as a measurement.

(4) a human capital level (EDU). The average years of education per province are used as a measurement [60]. The calculation formula is:

Average years of education = (Number of illiterate people + Number of people with primary education × 6 + Number of people with junior high education × 9 + Number of people with senior high and vocational education × 12 + Number of people with college education and above × 16) / Total population aged 6 and above.

(5) a full level of consumption (Con). The ratio of social retail sales to GDP is used as a measurement. This ratio reflects consumption levels, with a higher value indicating stronger consumption capacity.

Data description: This paper selects data with reference to Ren at al. [7] and other scholars. China has 34 provincial-level administrative regions; however, due to the severe lack of data for Taiwan Province, Tibet Autonomous Region, Hong Kong Special Administrative Region, and Macao Special Administrative Region, they are excluded. The impact of the pandemic is also fully considered, as it may distort certain values such as GDP, total social consumption, and per capita GDP. At the same time, to ensure the completeness of the data, this paper utilizes data spanning from 2011 to 2020 for 30 provinces (autonomous regions) in China. The original data are sourced from the "China Education Yearbook," "China Statistical Yearbook," "China Finance Yearbook," "China Environmental Statistics Yearbook," "Provincial Statistical Yearbooks," and the Wind database. For the very few missing data points, linear interpolation methods are used to fill in the gaps. The variables used in this paper and their descriptive statistics are shown in Tables 1 and 2.

**4.2.1. Analysis of descriptive statistics.** In the present study, the variables Con, EDU, FIX, PGDP, GOVER, DFI, ER and A were analyzed with descriptive statistics. Results are shown in Table 2.

**Table 1. Names of variables.**

| Nature of variables | Variable nameed industs | Measurement of variables | Notation |
|---|---|---|---|
| Explained variable | Advanced industrial structure | Index of advanced industrial structure | A |
| Explanatory variables | Digital inclusive finance | Digital inclusive finance index | DFI |
| | Environmental regulation | Environmental regulation intensity index | ER |
| Control variable | Level of government intervention | Fiscal expenditure/GDP | GOVER |
| | Level of economic development | GDP per capita | PGDP |
| | Level of fixed asset investment | Investment/GDP | FIX |
| | Human capital level | Average years of education | EDU |
| | Consumption level | Total retail consumption/GDP | Con |

## 4.3. Spatial correlation test for industrial structure upgrading

Moran's I index for the upgrading of China's industrial structure from 2011–2020 was computed using Stata 17.0 software, with the Z value determining its significance. The results are displayed in Table 3. The Moran's I values, derived from the global spatial autocorrelation test for this period, show little variation and are consistently positive. This implies a consistent positive spatial correlation in the industrial structure among provinces.

The regional spatial autocorrelation diagrams of industrial structure upgrading in 2011, 2015 and 2020 are drawn by stata17.0 software. The diagram reveals that most observations are predominantly located in both the first and third quadrants. This suggests that regions with high indices tend to cluster together, as do those with low indices. This pattern is primarily marked by instances of 'high-high clustering' and 'low-low clustering.' In other words, areas with advanced industrial structure upgrading are situated next to each other, and similarly, regions with lower levels of industrial upgrading are also spatially grouped together. Nevertheless, a small number of regions fall within the second and fourth quadrants based on their Moran values, signifying the presence of spatial disparities in the upgrading of industrial structures across China. All metrics associated with the enhancement of industrial structures are positive, indicating a consistent positive spatial relationship in the industrial framework among various provinces. In general, the spatial characteristics of "local convergence and overall divergence" are presented. Using ARGICS10.8 software to draw the LISA chart of the Moran index by year, we can observe the local spatial autocorrelation. In 2011 (Fig 2), the number of significant High-Low (H-L) type regions and significant Low-High (L-H) type regions is comparable. The H-L type regions are mainly in Guizhou and Guangdong, while the L-H type regions are in Inner Mongolia and Hebei, with the High-High (H-H) type area in Shandong. As the Moran value decreases in 2015 (Fig 3), Low-Low (L-L) type clustering

**Table 2. Descriptive statistics.**

| Variable | Number of Observations | Mean | Standard deviation | Minimum value | Maximum values |
|---|---|---|---|---|---|
| A | 300 | 6.703 | 0.304 | 6.175 | 7.657 |
| ER | 300 | 11.211 | 5.507 | 3.684 | 54.711 |
| DFII | 300 | 217.246 | 96.968 | 18.33 | 431.928 |
| ER*DFII | 300 | 2578.968 | 1836.67 | 84 | 11259.986 |
| GOVER | 300 | 0.264 | 0.114 | 0.12 | 0.758 |
| EDU | 300 | 1.245 | 0.198 | 0.753 | 1.717 |
| PGDP | 300 | 52635.291 | 26286.75 | 13119 | 164220 |
| FIX | 300 | 0.787 | 0.264 | 0.142 | 1.48 |
| Con | 300 | 0.398 | 0.062 | 0.222 | 0.61 |

**Table 3. Global spatial autocorrelation test result values.**

| Year | Moran, I | E(I) | Sd(I) | Z | P value |
|------|----------|------|-------|---|---------|
| 2011 | 0.6196 | -0.0345 | 0.0980 | 6.6714 | 0.0000 |
| 2012 | 0.6164 | -0.0345 | 0.0983 | 6.6197 | 0.0000 |
| 2013 | 0.6195 | -0.0345 | 0.0985 | 6.6397 | 0.0000 |
| 2014 | 0.6202 | -0.0345 | 0.0984 | 6.6542 | 0.0000 |
| 2015 | 0.5894 | -0.0345 | 0.0990 | 6.3023 | 0.0000 |
| 2016 | 0.5938 | -0.0345 | 0.0997 | 6.3033 | 0.0000 |
| 2017 | 0.5840 | -0.0345 | 0.0987 | 6.2697 | 0.0000 |
| 2018 | 0.5833 | -0.0345 | 0.0981 | 6.2987 | 0.0000 |
| 2019 | 0.6080 | -0.0345 | 0.0983 | 6.5365 | 0.0000 |
| 2020 | 0.6199 | -0.0345 | 0.0990 | 6.6097 | 0.0000 |

occurs, mainly in the Yunnan region. As the Moran value increases, the spatial correlation further increases, and in 2020 (Fig 4), there are significant H-H, L-L, and L-H type regions.

## 4.4. Empirical results and analysis

Table 4 displays the assessment of the chosen spatial econometric model for the data analyzed in this research. Based on the findings from the Hausman, Wald, LM, and LR tests, and following the framework suggested by Anselin and Florax [61], the fixed effects SDM has been determined to be the optimal model.

In Table 5, a comparative analysis between SDM and fixed effects panel regression using the full sample reveals broadly consistent conclusions. However, the spatial SDM exhibits a spatial autocorrelation coefficient (rho) that is significantly positive, indicating a positive spatial spillover effect in the upgrading of industrial structures. In contrast, conventional panel regression neglects the impact of geographical location on economic activities, potentially leading to biased model estimations. This finding also validates the rationale behind selecting a spatial econometric model for analysis in this study. The analysis revealed a notable positive relationship between ER and industrial structure upgrading, evident in both SDM and ordinary fixed panel data models. This indicates that stringent ER can substantially boost industrial upgrading. The mechanism involves enhancing the entry threshold for relevant enterprises, thereby creating 'industry barriers. Simultaneously, these regulations increase investments in

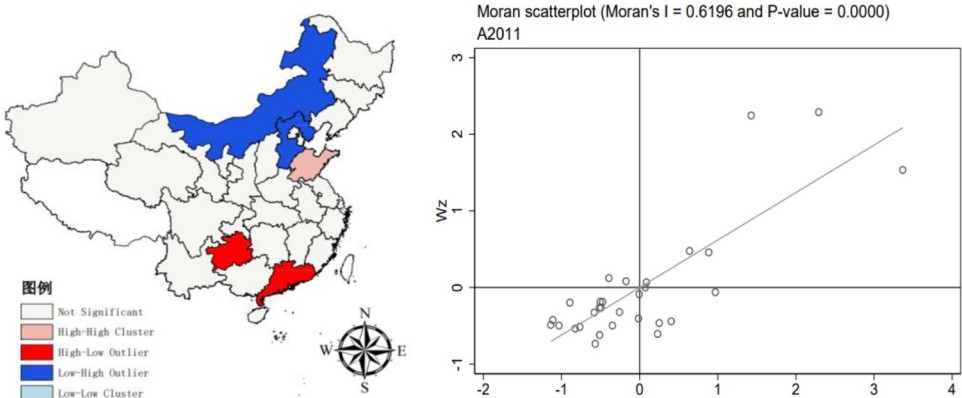

**Fig 2.** 2011 Moran scatterplot (left) and LISA plot (right). The underlying layer is quoted from the Sky Map website. Image is similar but not identical to the original. Therefore it is for illustrative purposes only.

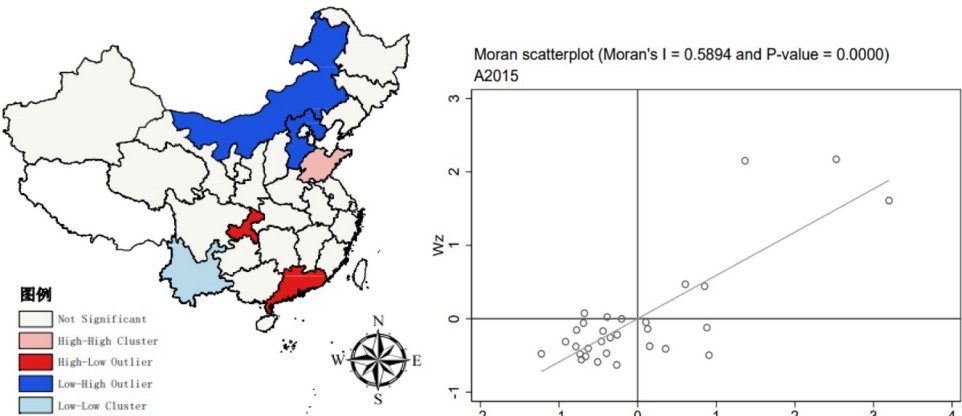

**Fig 3.** 2015 Moran scatterplot (left) and LISA plot (right). The underlying layer is quoted from the Sky Map website. Image is similar but not identical to the original. Therefore it is for illustrative purposes only.

pollution control, constraining the entry of new businesses and curbing the activities of enterprises with "three highs and one low". This has resulted in increasing costs, forcing reforms or closures, and eliminating outdated production capacities. Consequently, new factors of production shift towards emerging, efficient industrial sectors, optimizing the economic development model. Additionally, government subsidies to enterprises with innovative capacities reduce production costs and improve competitive advantages, directing social capital towards green industries and further enhancing industrial structure upgrading. DFI significantly contributes to industrial structure upgrading at the 1% level. This is mainly because DFI, with its features of "low threshold, high coverage, and high efficiency," is digitally empowered to match the needs of relevant green emerging enterprises accurately. It expedites the effective distribution of resources, suppresses financial exclusion due to misinformation, and effectively reduces the financial exclusion of MSMEs. By significantly lowering the financing barriers for SMEs, DFI expands their financing avenues, mitigating the challenges of obtaining financing and the high associated costs. However, the interaction terms of environmental regulation and DFI are significantly negatively correlated with industrial structure upgrading, indicating potential "mission drift". The aberrant development of finance and excessive speculation lead to "de-realization to virtualization" and other problems, which are detrimental to industrial structure

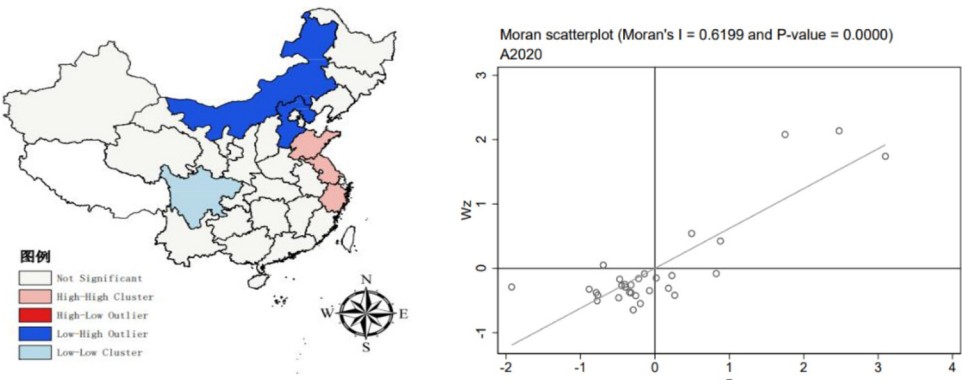

**Fig 4.** 2020 Moran scatterplot (left) and LISA plot (right). The underlying layer is quoted from the Sky Map website. Image is similar but not identical to the original. Therefore it is for illustrative purposes only.

**Table 4. Results of the spatial Durbin model applicability test.**

| Variable | Numerical Value | Variable | Numerical Value |
|---|---|---|---|
| LM- error | 13.068*** | R- LM- error | 0.159 |
| LM-lag | 17.190*** | R- LM-lag | 6.108** |
| LR-lrtest sdm sar | 47.61*** | LR- lrtest sdm sem | 70.77*** |
| wald -sdm | | 41.46*** | |
| Hausman | | 116.35*** | |

In the table*, **, and *** represent significance at the 10%, 5%, and 1% levels, respectively; numerical values represent standard errors; same below.

upgrading. A substitution dynamic exists between environmental regulation and DFI. In regions where DFI is highly developed, this relationship can encourage the growth of the secondary sector of industry instead of pivoting towards the tertiary industry, which typically flourishes under greater DFI. Conversely, in areas with stringent environmental regulations, the effectiveness of DFI is diminished, and its impact lessened.

The Spatial Durbin model is evaluated using Elhorst's [62] maximum likelihood (ML) approach. This method facilitates the estimation of the spatial influences of DFI and environmental regulation on industrial structure enhancement. The SDM model, accompanied by partial differential decomposition, is employed to assess both the immediate and consequential

**Table 5. SDM regression analysis and Fixed panel regression.**

| variables | SDM | | | | Fixed panel regression |
|---|---|---|---|---|---|
| | Main | Wx | Spatial | Variance | |
| ER | 0.0151*** | 0.0380*** | | | 0.0979*** |
| | (0.00382) | (0.0107) | | | (0.0120) |
| DFI | 0.00501*** | 0.00407** | | | 0.00376*** |
| | (0.000693) | (0.00197) | | | (0.000689) |
| ER* DFI | -6.68e-05*** | -0.000136*** | | | -0.000401*** |
| | (1.68e-05) | (4.30e-05) | | | (5.19e-05) |
| GOVER | 0.511*** | 0.509* | | | 1.914*** |
| | (0.0937) | (0.268) | | | (0.266) |
| EDU | 0.220*** | 0.351** | | | 2.211*** |
| | (0.0712) | (0.154) | | | (0.142) |
| PGDP | 4.04e-07 | 8.39e-06*** | | | 2.47e-05*** |
| | (1.28e-06) | (2.39e-06) | | | (1.58e-06) |
| FIX | -0.169*** | 0.0792 | | | -0.455*** |
| | (0.0459) | (0.146) | | | (0.149) |
| Con | -0.0471 | -0.871** | | | 4.016*** |
| | (0.134) | (0.354) | | | (0.387) |
| rho | 0.159* | | | | |
| | (0.0872) | | | | |
| sigma2_e | 0.0152*** | | | | |
| | (0.00124) | | | | |
| Observations | 300 | | | | |
| R² | 0.371 | | | | 0.995 |
| Number of id | 30 | | | | 30 |
| Individual Fixed | Yes | | | | Yes |
| Time Fixed | Yes | | | | Yes |

impacts (spatial spillover effects) of these variables. The findings presented in Table 6 indicate that regarding the direct effect, ER's influence on the upgrading of industrial structure is positively and significantly correlated at the 1% level. Additionally, ER's indirect effect demonstrates a significant positive spatial spillover, meaning that intensifying ER in one province positively impacts adjacent regions. This aligns with Hypothesis 2 proposed in this study. This may be due to the "Porter's hypothesis" mentioned in the previous section, the demonstration effect of environmental regulation, and other reasons. The influence of DFI on industrial structure enhancement is also positive and statistically significant at the 1% level. Concurrently, DFI exhibits a positive spatial spillover effect on industrial upgrading. In essence, if a province enhances its DFI framework, it will elevate the level of industrial structure advancement in surrounding areas, aligning with Hypothesis 1 presented in the study. This may be due to the innovative compensatory nature of DFI, the stimulation of consumption upgrading, and the larger regional coverage characteristics described above. Digital finance reduces the cost of enterprise financing, guides enterprises to adopt advanced technology and optimize resource allocation, and benefits the evolution of the regional economy towards a higher quality standard [63]. Additionally, the presence of DFI addresses the "long-tailed groups" in the financial market by providing financial services for these groups and increasing their accessibility. Consumption upgrading, and promote the optimization of industrial institutions. Under the indirect effect, the existence of DFI accelerates the construction of financial infrastructure between regions, strengthens the links between regions, breaks down the restrictions of regional barriers, accelerates the convenient and efficient flow of factors of production, and facilitates the upgrading of the industrial framework in adjacent areas. Nonetheless, the interaction term between ER and DFI is significantly negative at the 1% level. This suggests that the interplay between ER and DFI has a negative moderating effect. It impedes not only the upgrading of the industrial structure within the region but also proves detrimental to the industrial enhancement in neighboring regions, consistent with Hypothesis 4 outlined in the study. A likely explanation could be that when the degree of stringency in ER and DFI work simultaneously, the effect of "1+1>2" is not realized. ER requires enterprises to increase expenditure on pollution management. However, due to a lack of funds, DFI is insufficient to cover these costs. In cases of fund shortages, enterprises may have to divert funds from industrial upgrading and improving production efficiency. Excessive ER intensity can hinder industrial structure upgrading [64], causing funds to flee the region and leading relevant enterprises to relocate to adjacent areas. Additionally, the "mission drift" of DFI means it may not effectively help enterprises optimize the industrial chain and production equipment, boost production efficacy, or facilitate industrial structure upgrading. There is also a problem of "de-concentration"; DFI institutions may focus more on short-term interests and returns, neglecting long-term sustainable development considerations, ultimately leading to a negative regulatory effect on the region. The "competition at the bottom" strategy of government departments in neighboring regions leads to the transfer effect of ER, causing "three highs and one low" industries to move to neighboring regions, which obstructs the enhancement of the industrial structure in surrounding areas. This could also be due to local ER mandating businesses to adopt new environmental protection technologies and processes, potentially resulting in a negative regulatory impact on the region. Additionally, local ERs may require enterprises to introduce and apply new environmental protection technologies and techniques, which need an extended period to be transferred to neighboring regions before they can be effective. This "lagging effect" in time prevents short-term advancement of the industrial structure in adjacent areas.

**Table 6. SDM decomposition analysis.**

| Variables | Direct effects | Indirect effects | Total effects | Spatial | Variance |
|---|---|---|---|---|---|
| ER | 0.0165*** | 0.0478*** | 0.0643*** | | |
| | (0.00393) | (0.0120) | (0.0139) | | |
| DFI | 0.00513*** | 0.00564*** | 0.0108*** | | |
| | (0.000667) | (0.00198) | (0.00216) | | |
| ER* DFI | -7.11e-05*** | -0.000174*** | -0.000245*** | | |
| | (1.72e-05) | (4.81e-05) | (5.62e-05) | | |
| GOVER | 0.527*** | 0.693** | 1.220*** | | |
| | (0.0901) | (0.278) | (0.297) | | |
| EDU | 0.232*** | 0.447** | 0.679*** | | |
| | (0.0724) | (0.205) | (0.251) | | |
| PGDP | 7.59e-07 | 9.81e-06*** | 1.06e-05*** | | |
| | (1.26e-06) | (2.98e-06) | (3.05e-06) | | |
| FIX | -0.167*** | 0.0615 | -0.106 | | |
| | (0.0494) | (0.177) | (0.200) | | |
| Con | -0.0807 | -1.013** | -1.093** | | |
| | (0.126) | (0.434) | (0.472) | | |
| Rho | | | | 0.159* | |
| | | | | (0.0872) | |
| sigma2_e | | | | | 0.0152* |
| | | | | | (0.00124) |
| N | | | 300 | | |
| $R^2$ | | | 0.371 | | |
| Number of id | | | 30 | | |
| Individual Fixed | | | Yes | | |
| Time Fixed | | | Yes | | |

## 4.5. Regional heterogeneity analysis

Given the differences in the strength of environmental regulation and the maturity of DFI across the Eastern, Central, and Western regions, a sub-regional analysis is necessary.

Table 7 reveals that DFI significantly fosters the upgrading of industrial structures in the Eastern, Central, and Western regions at the 1% significance level, with an increasing influence coefficient in that order. This trend is due to the differing degrees of financial development in these regions, with the highest level in the East and lower levels in the Central and Western regions. DFI, supplementing traditional financial services, meets the financing needs in the Central and Western regions. DFI alleviates the financial exclusion resulting from asymmetric information. It effectively reduces the financial barriers for MSMEs, expands their access to finance, and enhances services such as third-party payments and digital insurance, thereby aiding in the improvement of the industrial structure Regarding ER, increased ER intensity in the Eastern region slightly hinders the advancement of the industrial structure. This is possibly attributed to the region's developed economic model and the proliferation of emerging industries, where excessive ER might act as an obstacle. In contrast, in the Central and Western regions, where these areas have historically served as 'pollution havens' due to the transfer of "three highs and one low" industries from the East, stronger ERs can aid in upgrading the industrial structure. However, this effect is only significantly evident in the Central and Western regions. From a sub-regional perspective, the moderating impacts of ER and DFI on the enhancing of industrial structures are not substantial across all regions.

**Table 7. Regional heterogeneity.**

| Variables | Eastern regions | Central Regions | Western regions | Central and western regions |
|---|---|---|---|---|
| ER | -0.000389 | 0.00390 | 0.0108 | 0.00517* |
| | (0.00333) | (0.00328) | (0.00837) | (0.00291) |
| DFI | 0.00143*** | 0.00161*** | 0.00186*** | 0.00134*** |
| | (0.000166) | (0.000382) | (0.000380) | (0.000236) |
| ER* DFI | 3.97e-06 | -6.27e-06 | -3.79e-05 | -9.22e-06 |
| | (1.20e-05) | (1.66e-05) | (3.55e-05) | (1.39e-05) |
| GOVER | 0.0650 | -2.052*** | 0.433 | 0.108 |
| | (0.289) | (0.673) | (0.338) | (0.282) |
| EDU | 0.253*** | 0.251 | 0.380*** | 0.329*** |
| | (0.0861) | (0.158) | (0.134) | (0.104) |
| PGDP | -2.58e-06*** | 2.34e-06 | -1.62e-06 | 1.43e-06 |
| | (5.29e-07) | (2.86e-06) | (2.28e-06) | (1.81e-06) |
| FIX | -0.107*** | 0.0973 | -0.0618 | 0.0106 |
| | (0.0359) | (0.0668) | (0.0629) | (0.0445) |
| Con | 0.0183 | 0.411** | 0.134 | 0.126 |
| | (0.0886) | (0.195) | (0.175) | (0.128) |
| C | 6.511*** | 6.039*** | 5.597*** | 5.693*** |
| | (0.137) | (0.264) | (0.280) | (0.192) |
| N | 110 | 80 | 110 | 190 |
| R2 | 0.986 | 0.876 | 0.850 | 0.842 |

## 4.6. Threshold effect analysis

Considering that there may be a nonlinear relationship between DFI and ER, this research utilizes Hansen's threshold regression model. It considers ER, DFI, and their interaction as threshold variables. The estimation is conducted using Stata17 through a self-sampling (Bootstrap) approach, involving 1,000 iterations for single-threshold, double-threshold, and triple-threshold scenarios respectively. The detailed outcomes presented in Table 8 indicate the presence of a single-threshold effect. This reveals an inverted 'U-shaped' correlation between DFI, ER, and the interaction of these two factors on the enhancement of industrial structures. Analyzing Tables 8 and 9, the threshold value for DFI is identified as 209.4549. As the intensity of DFI exceeds this threshold, the coefficient for industrial structure upgrading escalates from 0.0007589 to 0.0011347. For environmental regulation, the threshold is set at 10.4633. A

**Table 8. Threshold test results.**

| Threshold variables | Number of thresholds | F-statistics | P-values | 10% threshold value | 5% threshold value | 1% threshold value | Threshold value | 95% Period of confidence |
|---|---|---|---|---|---|---|---|---|
| DFII | Single Threshold | 40.89 | 0.0260 | 31.1079 | 36.8490 | 48.6472 | 209.4549 | 208.4611-209.9300 |
| | double threshold | 13.42 | 0.3110 | 20.3334 | 24.2829 | 32.1336 | | |
| ER | Single Threshold | 26.36 | 0.0280 | 20.3182 | 24.0269 | 31.9351 | 10.4633 | 10.3511-10.5288 |
| | double threshold | 3.89 | 0.8700 | 14.5853 | | | | |
| ER*DFII | Single Threshold | 35.55 | 0.0060 | 24.1939 | 27.9715 | 34.4028 | 1804.6507 | 1746.7536-1811.5098 |
| | double threshold | 10.36 | 0.4760 | 18.4809 | 21.7086 | 27.3821 | | |

**Table 9. Results of parameter estimation.**

| Threshold variables | Threshold value | 95%<br>Period of confidence | Threshold range | Ratio | T-value | P-value |
|---|---|---|---|---|---|---|
| DFII | 209.4549 | 208.4611- | DFII<209.4549 | 0.0007589 | 4.11 | 0.000 |
| | | 209.9300 | DFII≥209.4549 | 0.0011347 | 7.83 | 0.000 |
| ER | 10.4633 | 10.3511- | ER<10.4633 | -0.0055404 | -1.09 | 0.0278 |
| | | 10.5288 | ER≥10.4633 | -0.019739 | -2.87 | 0.005 |
| ER*DFII | 1804.6507 | 1746.7536 | ER*DFII<1804.6507 | 0.0003101 | 1.85 | 0.065 |
| | | 1811.5098 | | 0.0000166 | 0.44 | 0.663 |

heightened level of environmental regulation correlates with a reduced hindrance to industrial structure upgrading. Additionally, the interaction between environmental regulation and DFI demonstrates a single-threshold effect on industrial structure upgrading, characterized by an inverted 'U-shape' relationship. When the threshold value for environmental regulation is lower than 10.4633 and its intensity increases, the obstruction to industrial structure upgrading diminishes. However, when considering the interaction term of environmental regulation and DFI as the threshold variable, with a threshold value of 1804.6507, its effectiveness declines once this value is surpassed. This suggests an issue with the suitability of the interaction term between environmental regulation and DFI.

### 4.7. Robustness testing

To enhance the credibility and persuasiveness of the Spatial Durbin model's results, a robustness check is conducted. This analysis involves altering the core variables, control variables, and the weight matrix. In this study, following Li et al. [55], the rationalization of industrial structure is used as an explanatory variable. The weighting matrix was adjusted according to provincial distance for SDM model analysis. The majority of the outcomes align with the findings of the previous study, suggesting that the regression results are generally reliable. For detailed information, refer to Table 10.

## 5. Conclusions and recommendations

### 5.1. Conclusions

Utilizing the SDM to analyze panel data from 30 provinces in China for the years 2011–2020, this study empirically examines the effects of environmental regulation, digital inclusive finance, and their interaction on the upgrading of industrial structures. Panel regression analysis was employed to assess regional heterogeneity and threshold effects. The conclusions are summarized as follows:

Firstly, Moran's I for the industrial structure upgrade index indicates a positive spatial correlation, with the LISA map showing "high-high clustering" and "low-low clustering." Secondly, ER and DFI significantly promote the upgrade of the industrial structure both locally and in surrounding areas, under both the economic geography matrix and the provincial distance matrix, thereby facilitating high-quality industrial development. Thirdly, regional heterogeneity indicates that DFI promotes industrial structure upgrading in the eastern, central, and western regions, with the effect gradually strengthening. ER hinders industrial structure in the eastern region, suggesting a relatively well-balanced industrial structure. In contrast, ER promotes industrial structure upgrading in the central and western regions, improving the local industrial structure. The interaction term has a significant negative impact on industrial structure upgrading, highlighting the need for moderation when implementing both policies

**Table 10. Robustness test.**

| VARIABLES | Main | Wx | Spatial | Variance | Direct effects | Indirect effects | Total effects |
|---|---|---|---|---|---|---|---|
| ER | 1.775 | 71.03*** | | | 4.459 | 83.32*** | 87.78*** |
| | (7.804) | (17.61) | | | (7.978) | (20.27) | (21.87) |
| DFI | -0.000113 | 0.00448*** | | | 1.71e-05 | 0.00520** | 0.00522** |
| | (0.000617) | (0.00170) | | | (0.000600) | (0.00204) | (0.00217) |
| ER* DFI | -0.0447 | -0.267*** | | | -0.0536 | -0.321*** | -0.375*** |
| | (0.0407) | (0.0893) | | | (0.0418) | (0.103) | (0.116) |
| GOVER | 0.360*** | 0.0299 | | | 0.359*** | 0.101 | 0.460 |
| | (0.0946) | (0.248) | | | (0.0886) | (0.291) | (0.281) |
| EDU | -0.0950* | -0.160 | | | -0.101* | -0.206 | -0.307* |
| | (0.0547) | (0.127) | | | (0.0539) | (0.153) | (0.171) |
| PGDP | 3.69e-06*** | -5.74e-06*** | | | 3.54e-06*** | -6.02e-06** | -2.47e-06 |
| | (6.56e-07) | (1.98e-06) | | | (6.66e-07) | (2.43e-06) | (2.76e-06) |
| FIX | -0.0369 | -0.0171 | | | -0.0365 | -0.0289 | -0.0654 |
| | (0.0385) | (0.112) | | | (0.0400) | (0.130) | (0.146) |
| Con | -0.160 | 0.000251 | | | -0.166 | -0.0346 | -0.201 |
| | (0.115) | (0.275) | | | (0.111) | (0.309) | (0.349) |
| rho | 0.149* | | | | | | |
| | (0.0924) | | | | | | |
| sigma2_e | 0.0108*** | | | | | | |
| | (0.000888) | | | | | | |
| Observations | 300 | | | | | | |
| $R^2$ | 0.021 | | | | | | |
| Number of id | 30 | | | | | | |
| Individual Fixed | Yes | | | | | | |
| Time Fixed | Yes | | | | | | |

simultaneously. Fourthly, threshold tests reveal a single threshold effect, exhibiting an inverted "U" relationship, indicating that the dynamic relationship between ER, DFI, and their interaction towards the upgrade of the industrial structure is dynamic and subject to a degree of moderation.

### 5.2. Recommendations

According to the above analysis and the actual situation of China's economic development, policy recommendations are put forward.

**5.2.1. Increase the support of DFI and upgrade the scope of coverage.** Firstly, our approach will focus on further developing DFI to facilitate industrial upgrading in three key areas: breadth, depth, and digital support. Concurrently, this will speed up the movement of financial elements, enhance resource allocation efficiency, boost the impact of technological innovations, broaden service contexts, reduce corporate expenses, improve accessibility, diminish financial exclusion, support green industries, and expedite the process of industrial enhancement. Secondly, maintaining an appropriate level of ER to help into industrial structure upgrading. Thirdly, Governments and financial institutions can leverage digital technology to deliver financial services, particularly by offering financing support to SMEs as well as innovative businesses. Simultaneously, they can foster the growth of green industries through incentives, subsidies, and low-interest loans, thus aiding in the advancement of the industrial framework.

**5.2.2. Maintaining an appropriate level of ER to help upgrade industrial structure.**
Firstly, raising the entry thresholds of relevant enterprises, constructing "financial and technological barriers", increasing investment in pollution control, imposing production restrictions on enterprises with "three highs and one low", raising costs, and prompting enterprises to reform and eliminate. Secondly, we should formulate a flexible and adaptable ER strategy and increase the participation of enterprises in the formulation of regulations. The implementation of flexible regulatory strategies take into full account the economic benefits and affordability of enterprises, and increase the effectiveness of regulation. Involving enterprises in the decision-making process of environmental policies can increase the acceptance and effectiveness of regulation by enterprises and help policymakers better understand and balance the interests of all parties. Thirdly, consider the development of DFI and adjust the intensity of environmental regulatory measures according to the regional digital finance situation. ER can be moderately weakened in regions with better digital financial development, while it can be strengthened in regions with weaker digital financial development.

**5.2.3. Strengthen the regulation of DFI to avoid the problems of "mission drift" and "de-realization".**   Firstly, it should implement "penetrating regulation" of DFI, build an innovative financial regulatory system with the help of financial technology, realize comprehensive, real-time and effective regulation of digital inclusive finance, shape an all-encompassing financial regulatory system, strengthen the guidance of financial resources, improve the evaluation system, and promote the circulation and sharing of financial data. Secondly, strengthen information disclosure and require digital inclusive financial institutions to publicly and transparently disclose business and risk information in order to enhance public and market supervision. Thirdly, enhancing risk identification and management is crucial. Utilizing technologies like large-scale data analysis and AI technology can accelerate the precision and rapidity of risk detection, thereby preventing the accumulation and proliferation of risks.

# 6. Insights and shortcomings

## 6.1. Insights

**6.1.1. Strategic investment for sustainable development.**   The discovery of both inverted 'U-shaped' and 'U-shaped' relationships in threshold analysis, coupled with the spatial connections between ER, DFI, and the enhancement of industrial structures, indicates the importance of a dynamic and balanced investment approach. This approach must adapt to the differential impacts of DFI and ER through various phases of industrial progress. By integrating DFI appropriately with environmental strategies and acknowledging the critical thresholds where the effects of these elements change, a pathway towards sustainable and high-quality growth can be established.

**6.1.2. Tailored regional policies.**   The observed positive spatial correlation between the upgrading of the industrial structure and the varied effects of ER in different areas highlights the necessity for region-specific policy frameworks. These frameworks must consider the distinct economic and geographical attributes of the eastern, central, and western regions to formulate targeted strategies. These strategies should support industrial advancement in regions with favorable ERs and address the challenges in areas where these regulations hinder development.

## 6.2. Limitation and future research direction

Firstly, The study's data limitations include the fact that the data is only up to 2020, not accounting for recent years, particularly during the pandemic. Future researchers can use more accurate methods to collect data and include data from recent years, specifically

conducting separate regression analysis for the pandemic period, to make the conclusions of the article more robust.

Secondly, in addition to DFI and ER, China's industrial structure upgrading is influenced by other factors. The selection of control variables in this paper's model may be inadequate. Furthermore, although partial matrices were constructed for spatial regression, they may not fully capture the complex and diverse interrelationships between different regions. Relying solely on these two weight matrices may lead to an incomplete understanding of regional interactions. Future research can build weight matrices that encompass multiple dimensions, such as economic, social, and cultural factors, to more effectively and comprehensively reflect the actual conditions of each region.

## Acknowledgments

We thank the National Bureau of Statistics of China for providing open data to support the writing of this paper. Thanks are also due to OCHA Services for providing the map of China.

## Author Contributions

**Writing – original draft:** Liuwu Chen.

**Writing – review & editing:** Guimei Zhang.

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
