## [Decision Letter · Decision Letter 0]

21 Feb 2024

PONE-D-24-02477Digital Finance Inclusive, Environmental Regulation, and Industrial Structure Upgrade: A Spatial Durbin Model Analysis Based on Provincial Data in ChinaPLOS ONE

Dear Dr. Zhang,

Thank you for submitting your manuscript to PLOS ONE. After careful consideration, we feel that it has merit but does not fully meet PLOS ONE’s publication criteria as it currently stands. Therefore, we invite you to submit a revised version of the manuscript that addresses the points raised during the review process.

The submission requires further revisions towards both the theoretical and the empirical framework.

We look forward to receiving your revised manuscript.

Kind regards,

Stefan Cristian Gherghina, PhD. Habil.

Academic Editor

PLOS ONE

Journal Requirements:

3. PLOS requires an ORCID iD for the corresponding author in Editorial Manager on papers submitted after December 6th, 2016. Please ensure that you have an ORCID iD and that it is validated in Editorial Manager. To do this, go to ‘Update my Information’ (in the upper left-hand corner of the main menu), and click on the Fetch/Validate link next to the ORCID field. This will take you to the ORCID site and allow you to create a new iD or authenticate a pre-existing iD in Editorial Manager. Please see the following video for instructions on linking an ORCID iD to your Editorial Manager account: https://www.youtube.com/watch?v=_xcclfuvtxQ.

4. We note that Figures 2-4 in your submission contain [map/satellite] images which may be copyrighted. All PLOS content is published under the Creative Commons Attribution License (CC BY 4.0), which means that the manuscript, images, and Supporting Information files will be freely available online, and any third party is permitted to access, download, copy, distribute, and use these materials in any way, even commercially, with proper attribution. For these reasons, we cannot publish previously copyrighted maps or satellite images created using proprietary data, such as Google software (Google Maps, Street View, and Earth). For more information, see our copyright guidelines: http://journals.plos.org/plosone/s/licenses-and-copyright.

a. You may seek permission from the original copyright holder of Figures 2-4 to publish the content specifically under the CC BY 4.0 license. 

Additional Editor Comments (if provided):

The submission requires further revisions towards both the theoretical and the empirical framework.

Reviewers' comments:

Reviewer's Responses to Questions

**Comments to the Author**

1. Is the manuscript technically sound, and do the data support the conclusions?

Reviewer #1: Partly

Reviewer #2: Yes

Reviewer #3: No

2. Has the statistical analysis been performed appropriately and rigorously? 

Reviewer #1: Yes

Reviewer #2: No

Reviewer #3: No

3. Have the authors made all data underlying the findings in their manuscript fully available?

Reviewer #1: Yes

Reviewer #2: Yes

Reviewer #3: No

4. Is the manuscript presented in an intelligible fashion and written in standard English?

Reviewer #1: No

Reviewer #2: No

Reviewer #3: Yes

5. Review Comments to the Author

Reviewer #1: The authors from the results section never mention any implication for academics, stakeholders and practitioners. I suggest them to better address the results, the implications and the relevance of the study.

The attention to the language is not capillary. Some sentences are fragmented, and others are not properly written. I suggest the authors to make a professional English proofreading.

To address potential issues in the methodology related to spatial econometric modeling, three specific tests can be conducted. These tests aim to validate the model's assumptions, check for robustness, and ensure the reliability of the findings:

Test 1: Test for Spatial Autocorrelation

Purpose: To detect the presence of spatial autocorrelation in the residuals of the econometric model, which would indicate that the model has not fully captured spatial dependencies among regions.

Use Moran's I test on the residuals of the regression model. Moran's I is a measure of spatial autocorrelation and can indicate whether the residuals are randomly distributed across space or if there is a pattern that the model has failed to capture.

If significant spatial autocorrelation is found, it suggests that the model might need to incorporate additional spatial terms or use a different spatial weighting matrix.

Test 2: Robustness Checks Using Alternative Spatial Weight Matrices

Purpose: To assess how sensitive the model's results are to the choice of the spatial weighting matrix. This is crucial since the spatial weight matrix defines the structure of spatial interactions and dependencies among the units of analysis.

Re-estimate the model using alternative definitions of the spatial weighting matrix. For example, besides the economic geography matrix and the square of the road distance, consider matrices based on physical distance, shared borders, or even non-spatial relationships like economic similarity or historical trade links.

Compare the results across different matrices to check for consistency in the sign, magnitude, and significance of estimated parameters. Significant variations could indicate that the model's findings are highly sensitive to the choice of spatial matrix, necessitating further investigation.

Test 3: Cross-validation for Model Specification

Purpose: To validate the model specification and ensure that the chosen variables and their interactions robustly explain the industrial structure upgrading without overfitting the data.

Implement a cross-validation technique, such as k-fold cross-validation, where the dataset is divided into k subsets. The model is trained on k-1 subsets and tested on the remaining subset, and this process is repeated k times with each subset used once as the test set.

Evaluate the model's performance across different folds using metrics such as R-squared, mean squared error (MSE), or mean absolute error (MAE) for both training and validation sets. Consistent performance across folds suggests that the model is robust and not overfitting.

Reviewer #2: - The authors have not adequately delineated the criteria for sample selection in their methodology, leaving a significant gap in understanding the research framework.

- The authors have not sufficiently explained the study's implications or its innovative aspects, which limits the reader's understanding of its importance and original contribution to the finance literature.

two implications can be suggested:

Regional Policy Differentiation: The positive spatial correlation for industrial structure upgrading and the differential impacts of environmental regulation across regions imply that policy frameworks should be tailored regionally. Specifically, policies should account for the unique economic and geographical characteristics of eastern, central, and western regions. This approach would enable targeted strategies that enhance industrial upgrading in areas where environmental regulation is supportive and mitigate any inhibitory effects in regions where it is not.

Balanced Development and Investment Strategy: The identification of an inverted 'U-shaped' and a 'U-shaped' relationship in the threshold tests, alongside the spatial correlations between environmental regulation, direct foreign investment (DFI), and industrial structure upgrading, suggests the need for a balanced and dynamic investment strategy. Such a strategy should consider the varying effects of DFI and environmental regulation over different stages of industrial upgrading. Emphasizing the nuanced application of DFI in conjunction with environmental policies could foster a more sustainable and high-quality development, particularly by recognizing the threshold points where the impact of these factors shifts.

- The authors have not articulated the limitations of their study, an omission that restricts the scope of its interpretability and applicability in broader contexts.

- The study's contribution to the existing body of knowledge appears limited as it explores the relationship between environmental regulation, digital financial inclusion (DFI), and the upgrading of industrial structures—a topic already well-covered in the literature. The empirical findings, derived from panel data analysis across 30 Chinese provinces and employing the spatial Durbin model, reaffirm the established understanding that both DFI and environmental regulation can significantly influence industrial structure upgrades locally and in neighboring regions. Furthermore, the identification of a single-threshold effect characterized by an inverted 'U' relationship, along with the noted regional heterogeneity in the effects of DFI and environmental regulation, aligns with prior studies. These findings, while valuable, do not introduce novel insights into the dynamics between DFI, environmental regulation, and industrial upgrading but rather reinforce the recognized importance of these factors in promoting sustainable development. The study's emphasis on the need for a balanced approach to DFI and environmental regulation, to avoid mission drift and ensure effective industrial upgrading, underscores existing recommendations rather than offering new theoretical or methodological advancements.

Reviewer #3: The paper's contribution to the existing literature on environmental regulation and digital financial inclusion (DFI) in the context of industrial upgrading is limited, as it reiterates already well-established findings within the field. The methodology and results, while thorough, do not introduce new theoretical perspectives or empirical evidence that advances the understanding of these topics. The analysis aligns closely with previous studies, and the conclusions are consistent with established knowledge, resulting in a lack of innovative insights.

Introduction

• The introduction lacks a clear and concise research objective that directly states the study's primary aim, creating ambiguity regarding the specific problem the research intends to address within the broader context of digital finance inclusion, environmental regulation, and industrial structure upgrading.

Literature

• I recommend that the Author(s) restructure the literature review to better align with the paper's research objectives and to enhance its contribution to the field. This could involve a more focused examination of previous studies that directly relate to digital finance inclusion (DFI), environmental regulation, and industrial upgrading, particularly those employing spatial econometric models. Emphasizing research from highly ranked journals would strengthen the review’s foundation.

• The Author(s) should also critically assess the methodologies and findings of these studies, identifying gaps your research aims to address. Moreover, a discussion on the variable impacts of DFI and environmental regulation across different regions, especially within the context of China, would add depth.

• Concluding the section with a clear articulation of how the current study extends or fills gaps in existing literature could clarify its novel contribution. A conclusion paragraph in the literature review section is redundant.

Sample and Methodology

• The sample must be specified. This information is only provided in the abstract.

• To enhance the robustness of the findings and address potential concerns regarding the spatial econometric model's accuracy and applicability, the Author(s) could consider the following tests:

o Lagrange Multiplier (LM) Tests for Spatial Dependence: Conduct Lagrange Multiplier tests for both spatial lag and spatial error models to detect any hidden spatial dependence not captured by the initial model. This can help justify the choice of a spatial econometric model over traditional regression models.

o Cross-Validation with Non-Spatial Models: Compare the results from the spatial econometric model with those obtained from non-spatial econometric models, such as OLS or fixed-effects models, with and without spatially lagged independent variables. This comparison can validate the significance of incorporating spatial effects into the

Results

• The presentation of the results is unclear, and the discussion section fails to convey the study's implications and significance adequately.

Conclusion and Implications

• In the conclusion section, the Author(s) omits a discussion of the study's limitations and implications, which is essential for providing a comprehensive understanding of the research's scope, applicability, and potential impact on the field.

6. PLOS authors have the option to publish the peer review history of their article (what does this mean?). If published, this will include your full peer review and any attached files.

Reviewer #1: No

Reviewer #2: No

Reviewer #3: No

---

## [Author Response · Author response to Decision Letter 0]

16 Jul 2024

Dear Professor Habil.,

Glad to receive your E-mail for informing me that our paper entitled “Digital Finance Inclusive, Environmental Regulation, and Industrial Structure Upgrade: A Spatial Durbin Model Analysis Based on Provincial Data in China” (No: [PONE-D-24-02477]) needs major revisions and clarification before considering for publishing in PLOS ONE. Thanks for your help and the helpful comments and suggestions from the reviewers. The manuscript has been thoroughly revised according to the reviewers’ comments, and the answers to the comments are listed below: 

Reviewer #1:

Comment 1: The authors from the results section never mention any implication for academics, stakeholders and practitioners. I suggest them to better address the results, the implications and the relevance of the study.

Author’s response: Thank the reviewer for the comment. We have added the relevant content in the revised the manuscript (See Lines 947-958).

Comment 2: The attention to the language is not capillary. Some sentences are fragmented, and others are not properly written. I suggest the authors to make a professional English proofreading.

Author’s response: Thank the reviewer for the comment. We have revised the whole manuscript carefully and tried to avoid any grammar or syntax error. In addition, we had invited native English speaker to review this manuscript, and avoided any mistakes of tense, grammar and syntax in the new version. We believe that the language is now acceptable for the review process. 

Comment 3: To address potential issues in the methodology related to spatial econometric modeling, three specific tests can be conducted. These tests aim to validate the model's assumptions, check for robustness, and ensure the reliability of the findings:

Test 1: Test for Spatial Autocorrelation

Purpose: To detect the presence of spatial autocorrelation in the residuals of the econometric model, which would indicate that the model has not fully captured spatial dependencies among regions.

Use Moran's I test on the residuals of the regression model. Moran's I is a measure of spatial autocorrelation and can indicate whether the residuals are randomly distributed across space or if there is a pattern that the model has failed to capture.

If significant spatial autocorrelation is found, it suggests that the model might need to incorporate additional spatial terms or use a different spatial weighting matrix.

Author’s response: Thank the reviewer for the comment. To support our methodology, we have referenced numerous papers that have utilized the Moran I test, published in journals such as PLOS ONE. Additionally, we have conducted the Moran I test as part of our analysis. (See Line 636 and Table 3 Global spatial autocorrelation test result values).

Comment 4: Test 2: Robustness Checks Using Alternative Spatial Weight Matrices

Purpose: To assess how sensitive the model's results are to the choice of the spatial weighting matrix. This is crucial since the spatial weight matrix defines the structure of spatial interactions and dependencies among the units of analysis.

Re-estimate the model using alternative definitions of the spatial weighting matrix. For example, besides the economic geography matrix and the square of the road distance, consider matrices based on physical distance, shared borders, or even non-spatial relationships like economic similarity or historical trade links.

Compare the results across different matrices to check for consistency in the sign, magnitude, and significance of estimated parameters. Significant variations could indicate that the model's findings are highly sensitive to the choice of spatial matrix, necessitating further investigation.

Author’s response: Thank the reviewer for the comment. I have added the use of provincial distance matrices as a robustness test. (See Lines 840). Other matrices have not been explored in this paper and remain one of the outstanding issues to be addressed in the article.

Comment 5: Test 3: Cross-validation for Model Specification

Purpose: To validate the model specification and ensure that the chosen variables and their interactions robustly explain the industrial structure upgrading without overfitting the data.

Implement a cross-validation technique, such as k-fold cross-validation, where the dataset is divided into k subsets. The model is trained on k-1 subsets and tested on the remaining subset, and this process is repeated k times with each subset used once as the test set.

Evaluate the model's performance across different folds using metrics such as R-squared, mean squared error (MSE), or mean absolute error (MAE) for both training and validation sets. Consistent performance across folds suggests that the model is robust and not overfitting.

Author’s response: Thank the reviewer for the comment. We have added the Cross-validation for Model Specification (E* DFI). (See Lines 844, Table 10 Robustness test).

Reviewer #2: 

Comment 1: The authors have not adequately delineated the criteria for sample selection in their methodology, leaving a significant gap in understanding the research framework.

Author’s response: Thanks so much for the comments. I have fully articulated the criteria for sample selection in the methodology of the revised draft. (See Lines 614-623). 

Comment 2: The authors have not sufficiently explained the study's implications or its innovative aspects, which limits the reader's understanding of its importance and original contribution to the finance literature.

Author’s response: Thank the reviewer for the comment. We have revised the research significance and innovation of the article. (See Lines 94-102). We also add the conclusion of the article. (See Lines 911-929). 

Comment 3: - The authors have not articulated the limitations of their study, an omission that restricts the scope of its interpretability and applicability in broader contexts..

Author’s response: Thank the reviewer for the comment. We have added the limitations of the study. (See Lines 930-946)

Comment 4: The study's contribution to the existing body of knowledge appears limited as it explores the relationship between environmental regulation, digital financial inclusion (DFI), and the upgrading of industrial structures—a topic already well-covered in the literature. The empirical findings, derived from panel data analysis across 30 Chinese provinces and employing the spatial Durbin model, reaffirm the established understanding that both DFI and environmental regulation can significantly influence industrial structure upgrades locally and in neighboring regions. Furthermore, the identification of a single-threshold effect characterized by an inverted 'U' relationship, along with the noted regional heterogeneity in the effects of DFI and environmental regulation, aligns with prior studies. These findings, while valuable, do not introduce novel insights into the dynamics between DFI, environmental regulation, and industrial upgrading but rather reinforce the recognized importance of these factors in promoting sustainable development. The study's emphasis on the need for a balanced approach to DFI and environmental regulation, to avoid mission drift and ensure effective industrial upgrading, underscores existing recommendations rather than offering new theoretical or methodological advancements.

Author’s response: Thank the reviewer for the comment. There is innovation in our article. In particular, the mechanism of action part of the article is strictly based on the theoretical analysis of spatial direct and indirect effects. In addition, the article analyzes the impact of the interaction terms of DFI and environmental regulation on industrial structure upgrading.

Reviewer #3: 

Comment 1: The paper's contribution to the existing literature on environmental regulation and digital financial inclusion (DFI) in the context of industrial upgrading is limited, as it reiterates already well-established findings within the field. The methodology and results, while thorough, do not introduce new theoretical perspectives or empirical evidence that advances the understanding of these topics. The analysis aligns closely with previous studies, and the conclusions are consistent with established knowledge, resulting in a lack of innovative insights.

Author’s response: Thank the reviewer for the comment. There is innovation in our article. In particular, the mechanism of action part of the article is strictly based on the theoretical analysis of spatial direct and indirect effects. In addition, the article analyzes the impact of the interaction terms of DFI and environmental regulation on industrial structure upgrading.

Comment 2: Introduction. The introduction lacks a clear and concise research objective that directly states the study's primary aim, creating ambiguity regarding the specific problem the research intends to address within the broader context of digital finance inclusion, environmental regulation, and industrial structure upgrading.

Author’s response: Thank the reviewer for the comment. We have rewritten the introduction section. (See Lines47-93)

Comment 3: I recommend that the Author(s) restructure the literature review to better align with the paper's research objectives and to enhance its contribution to the field. This could involve a more focused examination of previous studies that directly relate to digital finance inclusion (DFI), environmental regulation, and industrial upgrading, particularly those employing spatial econometric models. Emphasizing research from highly ranked journals would strengthen the review’s foundation.

Author’s response: Thank the reviewer for the comment. We have moderately adjusted the literature review by adding relevant literature. (See Lines108-114;157-158;162-164;166-178;200-203;214-216)However, there are still fewer studies directly related to digital financial inclusion (DFI), environmental regulation and industrial upgrading in the first place. 

Comment 4: The Author(s) should also critically assess the methodologies and findings of these studies, identifying gaps your research aims to address. Moreover, a discussion on the variable impacts of DFI and environmental regulation across different regions, especially within the context of China, would add depth.

Author’s response: Thank the reviewer for the comment. We have added reviews of the literature.

Comment 5: Concluding the section with a clear articulation of how the current study extends or fills gaps in existing literature could clarify its novel contribution. A conclusion paragraph in the literature review section is redundant.

Author’s response: Thank the reviewer for the comment. We have revised the literature innovations. (See Lines 217-225).And ,we deleted some of them.

Comment 6: Sample and Methodology. The sample must be specified. This information is only provided in the abstract.

Author’s response: Thank the reviewer for the comment. W We have supplemented the sample description section. (See Lines 614-623).

Comment 7: To enhance the robustness of the findings and address potential concerns regarding the spatial econometric model's accuracy and applicability, the Author(s) could consider the following tests: Lagrange Multiplier (LM) Tests for Spatial Dependence: Conduct Lagrange Multiplier tests for both spatial lag and spatial error models to detect any hidden spatial dependence not captured by the initial model. This can help justify the choice of a spatial econometric model over traditional regression models.

Author’s response: Thank the reviewer for the comment. We have already done the relevant tests in the article. (See Lines 688 Table 4 Results of the spatial Durbin model applicability test).

Comment 8:Cross-Validation with Non-Spatial Models: Compare the results from the spatial econometric model with those obtained from non-spatial econometric models, such as OLS or fixed-effects models, with and without spatially lagged independent variables. This comparison can validate the significance of incorporating spatial effects into the.

Author’s response: The choice of the spatial Durbin model is the most appropriate due to the tests that have been done earlier in this paper. So both spatial lag and spatial error models are not listed in the writing of this paper.

Comment 9:Results. The presentation of the results is unclear, and the discussion section fails to convey the study's implications and significance adequately.

Author’s response: Thank the reviewer for the comment. We have revised the conclusion section of the article. (See Lines 847-864). We've also added a revelation section to the article. (See Lines 911-929).

Comment 10: Conclusion and Implications. In the conclusion section, the Author(s) omits a discussion of the study's limitations and implications, which is essential for providing a comprehensive understanding of the research's scope, applicability, and potential impact on the field.

Author’s response: Thank the reviewer for the comment. We have added a limitations section to the article. (See Lines 931-946).

We have reviewed point by point in the new revised version, marked by red character, according to your comments. I hope the above answers and revisions are satisfactory, and the paper will be acceptable for publication in PLOS ONE.

---

## [Decision Letter · Decision Letter 1]

25 Jul 2024

PONE-D-24-02477R1Digital Finance Inclusive, Environmental Regulation, and Industrial Structure Upgrade: A Spatial Durbin Model Analysis Based on Provincial Data in ChinaPLOS ONE

Dear Dr. Zhang,

Thank you for submitting your manuscript to PLOS ONE. After careful consideration, we feel that it has merit but does not fully meet PLOS ONE’s publication criteria as it currently stands. Therefore, we invite you to submit a revised version of the manuscript that addresses the points raised during the review process.

We look forward to receiving your revised manuscript.

Kind regards,

Stefan Cristian Gherghina, PhD. Habil.

Academic Editor

PLOS ONE

**Additional Editor Comments:**

The manuscript still requires further revision towards the introductory section, as well as the methodological approach. As well, the quality of the English language should be improved.

Reviewers' comments:

Reviewer's Responses to Questions

**Comments to the Author**

1. If the authors have adequately addressed your comments raised in a previous round of review and you feel that this manuscript is now acceptable for publication, you may indicate that here to bypass the “Comments to the Author” section, enter your conflict of interest statement in the “Confidential to Editor” section, and submit your "Accept" recommendation.

Reviewer #1: (No Response)

Reviewer #2: (No Response)

Reviewer #3: (No Response)

2. Is the manuscript technically sound, and do the data support the conclusions?

Reviewer #1: Yes

Reviewer #2: Yes

Reviewer #3: Partly

3. Has the statistical analysis been performed appropriately and rigorously? 

Reviewer #1: Yes

Reviewer #2: Yes

Reviewer #3: No

4. Have the authors made all data underlying the findings in their manuscript fully available?

Reviewer #1: No

Reviewer #2: Yes

Reviewer #3: Yes

5. Is the manuscript presented in an intelligible fashion and written in standard English?

Reviewer #1: Yes

Reviewer #2: Yes

Reviewer #3: No

6. Review Comments to the Author

**Reviewer #1:** Major Revisions

The introduction should provide a clearer and more concise research objective. This will help eliminate any ambiguity regarding the specific problem the study intends to address within the context of digital finance inclusion, environmental regulation, and industrial structure upgrading.

Additionally, the methodology section needs a detailed explanation of the criteria for sample selection. This will strengthen the research framework and ensure a better understanding of the study's context. Furthermore, the implications and innovative aspects of the study should be elaborated upon to highlight its importance and original contributions to the finance literature.

This includes discussing the study's limitations, as this is crucial for understanding the scope and applicability of the research in broader contexts. Lastly, ensure that the presentation of results and the discussion sections clearly convey the study's significance and implications, enhancing the overall comprehensibility of the manuscript.

Minor Revisions

The language throughout the manuscript needs careful proofreading to address fragmented sentences and improve overall readability. Professional English proofreading is recommended to avoid any grammatical or syntactical errors.

Additionally, the literature review should be better aligned with the paper's research objectives, including a focused examination of previous studies related to digital finance inclusion, environmental regulation, and industrial upgrading, especially those employing spatial econometric models.

Emphasizing research from highly ranked journals will strengthen the foundation of the literature review.

Finally, a more critical assessment of the methodologies and findings of these studies should be included, identifying gaps that your research aims to address and articulating how the current study extends or fills these gaps, thereby clarifying its novel contributions.

**Reviewer #2:** I appreciate your diligent efforts made in revising your manuscript. I am pleased to observe that all of my previous comments and suggestions have been thoroughly addressed. The comprehensive manner in which you have incorporated these changes is commendable, and it is evident that the quality and clarity of the manuscript have been significantly enhanced as a result.

The revised version of your paper now aligns well with the high standards expected by the journal, reflecting both the robustness of your research and the depth of your analysis. Your commitment to improving the manuscript is truly admirable.

Congratulation!

**Reviewer #3: **I appreciate your efforts in revising the manuscript. All my previous suggestions and requests have

been adequately implemented in the revised version of the manuscript.

However, I have a few further recommendations that I believe will further enhance the quality and clarity of your work.

1) The title of the manuscript could be streamlined and made more explanatory. According to

Ecarnot et al. (2015), “The title should contain keywords to reflect the main issues in your

article. It should also awaken the potential reader’s interest and incite the desire to read

your work in full”.

2) The abstract could be improved; the current version does not highlight the study's relevance,

novelty, and contribution. To enhance its clarity and impact, please following these

suggestions:

- Conciseness and clarity: Simplify sentences to enhance readability and ensure that each

sentence conveys a clear, single idea.

- Logical flow: Ensure that the abstract follows a logical progression from the introduction of

the problem through methodology to key findings and implications.

- Focus on key results: Emphasize the most significant findings and their implications rather

than providing extensive details on regional differences or specific effects.

3) The keywords should be refined in order to accurately represent the topic developed in your

study and ensure coherence with the JEL classification.

4) It is essential to conduct a thorough proofreading of the manuscript. This will help refine the

constructs and enhance the overall quality of the writing.

5) Given the high number of tables and figures, it is recommended that detailed captions be

provided for tables and figures. The captions should provide a clear and detailed explanation

of the content or representation of each table or figure, thus facilitating the reader's

understanding of the data presented.

6) Please ensure that every table and figure is appropriately referenced within the manuscript.

I am confident these additional revisions will contribute to the manuscript's clarity and impact.

7. PLOS authors have the option to publish the peer review history of their article (what does this mean?). If published, this will include your full peer review and any attached files.

Reviewer #1: No

Reviewer #2: No

Reviewer #3: No

---

## [Author Response · Author response to Decision Letter 1]

21 Aug 2024

Additional Editor Comments:

The manuscript still requires further revision towards the introductory section, as well as the methodological approach. As well, the quality of the English language should be improved.

Author’s response: Thank the editor for the comment. We have revised the introduction and methodology sections as required, and we have also thoroughly checked the entire text. (See Lines 45-54;76-85;94-96;557-587)

Reviewer #1:

Comment 1: The introduction should provide a clearer and more concise research objective. This will help eliminate any ambiguity regarding the specific problem the study intends to address within the context of digital finance inclusion, environmental regulation, and industrial structure upgrading.

Author’s response: Thank the reviewer for the comment. We have revised the introduction and rewritten some parts, making the issues clearer. (See Lines 45-54;76-85;94-96).

Comment 2: Additionally, the methodology section needs a detailed explanation of the criteria for sample selection. This will strengthen the research framework and ensure a better understanding of the study's context. 

Author’s response: Thank the reviewer for the comment. We have provided additional explanations and made the relevant revisions. (See Lines 557-574;579-581)

Comment 3: Furthermore, the implications and innovative aspects of the study should be elaborated upon to highlight its importance and original contributions to the finance literature.

Author’s response: Thank the reviewer for the comment. We have summarized the innovative points of the article. (See Lines 86-96)

Comment 4: This includes discussing the study's limitations, as this is crucial for understanding the scope and applicability of the research in broader contexts. Lastly, ensure that the presentation of results and the discussion sections clearly convey the study's significance and implications, enhancing the overall comprehensibility of the manuscript

Author’s response: Thank the reviewer for the comment. We have supplemented and revised this section accordingly. (See Lines 885-899). We have also rewritten the conclusion. (See Lines 806-821).

Comment 5:The language throughout the manuscript needs careful proofreading to address fragmented sentences and improve overall readability. Professional English proofreading is recommended to avoid any grammatical or syntactical errors.

Author’s response: Thank the reviewer for the comment. We have revised the whole manuscript carefully and tried to avoid any grammar or syntax error. In addition, we had invited native English speaker to review this manuscript, and avoided any mistakes of tense, grammar and syntax in the new version. We believe that the language is now acceptable for the review process. 

Comment 6: Additionally, the literature review should be better aligned with the paper's research objectives, including a focused examination of previous studies related to digital finance inclusion, environmental regulation, and industrial upgrading, especially those employing spatial econometric models. Emphasizing research from highly ranked journals will strengthen the foundation of the literature review.

Author’s response: Thank the reviewer for the comment. We have revised parts of the literature review and included articles with a high impact on spatial econometrics related to the article's theme. (See Lines 111-114;159-168;185-190;202-205).

Comment 7: Finally, a more critical assessment of the methodologies and findings of these studies should be included, identifying gaps that your research aims to address and articulating how the current study extends or fills these gaps, thereby clarifying its novel contributions.

Author’s response: We have further verified the article. The empirical results effectively support the hypotheses and include robust regression analyses.

Reviewer #2: 

Comment 1: I appreciate your diligent efforts made in revising your manuscript. I am pleased to observe that all of my previous comments and suggestions have been thoroughly addressed. The comprehensive manner in which you have incorporated these changes is commendable, and it is evident that the quality and clarity of the manuscript have been significantly enhanced as a result. The revised version of your paper now aligns well with the high standards expected by the journal, reflecting both the robustness of your research and the depth of your analysis. Your commitment to improving the manuscript is truly admirable.

Congratulation!

Author’s response: Thanks so much for the comments. Thank you for your guidance.

Reviewer #3: 

Comment 1: 1) The title of the manuscript could be streamlined and made more explanatory. According to Ecarnot et al. (2015), “The title should contain keywords to reflect the main issues in your article. It should also awaken the potential reader’s interest and incite the desire to read your work in full”.

Author’s response: Thank the reviewer for the comment. We have revised the title according to your request. (See Lines1-3)

Comment 2: The abstract could be improved; the current version does not highlight the study's relevance, novelty, and contribution. To enhance its clarity and impact, please following these suggestions:

- Conciseness and clarity: Simplify sentences to enhance readability and ensure that each

sentence conveys a clear, single idea.

- Logical flow: Ensure that the abstract follows a logical progression from the introduction of the problem through methodology to key findings and implications.

- Focus on key results: Emphasize the most significant findings and their implications rather than providing extensive details on regional differences or specific effects.

Author’s response: Thank the reviewer for the comment. We have rewritten the abstract section. (See Lines23-40)

Comment 3: The keywords should be refined in order to accurately represent the topic developed in your study and ensure coherence with the JEL classification.

Author’s response: Thank the reviewer for the comment. We have made the necessary revisions as requested. (See Lines41-42)

Comment 4: It is essential to conduct a thorough proofreading of the manuscript. This will help refine the constructs and enhance the overall quality of the writing.

Author’s response: Thank the reviewer for the comment. We have revised the whole manuscript carefully and tried to avoid any grammar or syntax error. In addition, we had invited native English speaker to review this manuscript, and avoided any mistakes of tense, grammar and syntax in the new version. We believe that the language is now acceptable for the review process. 

Comment 5: Given the high number of tables and figures, it is recommended that detailed captions be provided for tables and figures. The captions should provide a clear and detailed explanation of the content or representation of each table or figure, thus facilitating the reader's understanding of the data presented.

Author’s response: Thank the reviewer for the comment. We have made the relevant revisions as requested. (See Lines 505-509;586-587;).

Comment 6: Please ensure that every table and figure is appropriately referenced within the manuscript.

Author’s response: Thank the reviewer for the comment. 

Author’s response: Thank the reviewer for the comment. We have revised the fields that did not use the relevant table contents.

---

## [Decision Letter · Decision Letter 2]

6 Sep 2024

Research on the Impact of Digital Financial Inclusion and Environmental Regulation on Industrial Structure Upgrading-Based on Spatial Durbin Model Analysis

PONE-D-24-02477R2

Dear Dr. Zhang,

We’re pleased to inform you that your manuscript has been judged scientifically suitable for publication and will be formally accepted for publication once it meets all outstanding technical requirements.

Kind regards,

Dipendra Karki, Ph.D.

Academic Editor

PLOS ONE

Additional Editor Comments (optional):

Dear Prof. Guimei Zhang,

Thank you for your revised submission of the manuscript titled "Research on the Impact of Digital Financial Inclusion and Environmental Regulation on Industrial Structure Upgrading-Based on Spatial Durbin Model Analysis." I have carefully reviewed your manuscript, along with the detailed comments provided by the reviewers, and I would like to acknowledge the substantial improvements made in response to the initial feedback.

While the manuscript has undergone significant improvement, I recommend a final round of proofreading to address any minor language issues, as highlighted by Reviewer #1 and Reviewer #3. Also, please ensure that all tables and figures are fully referenced in the text and accompanied by detailed captions for clarity.

Your efforts in addressing the reviewers' comments are commendable, and I believe your study makes a valuable contribution to the ongoing research in digital financial inclusion and its relationship with industrial upgrading. With a few minor revisions, the manuscript is now suitable for publication in PLOS ONE.

Thank you once again for your hard work and dedication in improving the quality of the manuscript.

Best regards,

Dr. Dipendra Karki

Reviewers' comments:

Reviewer's Responses to Questions

**Comments to the Author**

1. If the authors have adequately addressed your comments raised in a previous round of review and you feel that this manuscript is now acceptable for publication, you may indicate that here to bypass the “Comments to the Author” section, enter your conflict of interest statement in the “Confidential to Editor” section, and submit your "Accept" recommendation.

Reviewer #1: All comments have been addressed

Reviewer #2: All comments have been addressed

Reviewer #3: All comments have been addressed

2. Is the manuscript technically sound, and do the data support the conclusions?

Reviewer #1: Yes

Reviewer #2: Yes

Reviewer #3: Yes

3. Has the statistical analysis been performed appropriately and rigorously? 

Reviewer #1: Yes

Reviewer #2: Yes

Reviewer #3: Yes

4. Have the authors made all data underlying the findings in their manuscript fully available?

Reviewer #1: Yes

Reviewer #2: Yes

Reviewer #3: No

5. Is the manuscript presented in an intelligible fashion and written in standard English?

Reviewer #1: Yes

Reviewer #2: Yes

Reviewer #3: Yes

6. Review Comments to the Author

Reviewer #1: The revisions made by the authors have addressed all the concerns raised in the initial review. The introduction is now clearer, with well-defined research objectives, and the methodology section provides a more detailed explanation of the sample selection criteria. The literature review has been appropriately aligned with the study's objectives, incorporating relevant and high-impact studies. The language throughout the manuscript has been significantly improved, and the overall clarity and quality of the manuscript have been enhanced. The manuscript is now suitable for consideration in its current form.

Reviewer #2: The authors have successfully implemented all the suggestions provided in the previous round of reviews. The revisions have resulted in a clearer and more coherent manuscript. The improvements to the introduction, methodology, and language have strengthened the paper. I am satisfied with the changes made and believe the manuscript is now ready for publication.

Reviewer #3: The Author(s) have thoroughly addressed all the comments and suggestions I indicated in my previous revision.

As requested, the Author(s) have improved the abstract; the revised version more effectively emphasizes the study's aim and contributions.

The keywords have been refined, and detailed captions have been added to the tables and figures for greater clarity. The manuscript has been subjected to rigorous proofreading, resulting in enhanced readability and overall quality.

Given the comments, I find the revised manuscript acceptable for publication.

7. PLOS authors have the option to publish the peer review history of their article (what does this mean?). If published, this will include your full peer review and any attached files.

Reviewer #1: No

Reviewer #2: No

Reviewer #3: No

---

## [Editor Report · Acceptance letter]

20 Sep 2024

PONE-D-24-02477R2 

PLOS ONE

Dear Dr. Zhang, 

I'm pleased to inform you that your manuscript has been deemed suitable for publication in PLOS ONE. Congratulations! Your manuscript is now being handed over to our production team.

Kind regards, 

on behalf of

Dr. Dipendra Karki 

Academic Editor

PLOS ONE